# Structural basis for SARS-CoV-2 Delta variant recognition of ACE2 receptor and broadly neutralizing antibodies

Yifan Wang [1,2,4], Caixuan Liu[1,2,4], Chao Zhang [3,4], Yanxing Wang [1,4], Qin Hong[1,2,4], Shiqi Xu [3], Zuyang Li[1,2], Yong Yang[3], Zhong Huang [3✉] & Yao Cong [1,2✉]

The SARS-CoV-2 Delta variant is currently the dominant circulating strain in the world. Uncovering the structural basis of the enhanced transmission and altered immune sensitivity of Delta is particularly important. Here we present cryo-EM structures revealing two conformational states of Delta spike and S/ACE2 complex in four states. Our cryo-EM analysis suggests that RBD destabilizations lead to population shift towards the more RBD-up and S1 destabilized fusion-prone state, beneficial for engagement with ACE2 and shedding of S1. Noteworthy, we find the Delta T478K substitution plays a vital role in stabilizing and reshaping the RBM loop[473-490], enhancing interaction with ACE2. Collectively, increased propensity for more RBD-up states and the affinity-enhancing T478K substitution together contribute to increased ACE2 binding, providing structural basis of rapid spread of Delta. Moreover, we identify a previously generated MAb 8D3 as a cross-variant broadly neutralizing antibody and reveal that 8D3 binding induces a large K478 side-chain orientation change, suggesting 8D3 may use an "induced-fit" mechanism to tolerate Delta T478K mutation. We also find that all five RBD-targeting MAbs tested remain effective on Delta, suggesting that Delta well preserves the neutralizing antigenic landscape in RBD. Our findings shed new lights on the pathogenicity and antibody neutralization of Delta.

[1] State Key Laboratory of Molecular Biology, National Center for Protein Science Shanghai, Shanghai Institute of Biochemistry and Cell Biology, Center for Excellence in Molecular Cell Science, Chinese Academy of Sciences, Shanghai, China. [2] University of Chinese Academy of Sciences, Beijing, China. [3] CAS Key Laboratory of Molecular Virology and Immunology, Institut Pasteur of Shanghai, Chinese Academy of Sciences, University of Chinese Academy of Sciences, Shanghai, China. [4] These authors contributed equally: Yifan Wang, Caixuan Liu, Chao Zhang, Yanxing Wang, Qin Hong. ✉email: huangzhong@ips.ac.cn; cong@sibcb.ac.cn

Severe acute respiratory syndrome coronavirus 2 (SARS-CoV-2) is the causative agent of coronavirus disease 2019 (COVID-19) that has been affecting the whole world since December 2019. The spike (S) protein of SARS-CoV-2 is a glycoprotein that forms trimers protruding from the virion surface and mediates viral entry into cells[1–5]. The S trimer uses its receptor-binding domain (RBD) to bind with the host-cell receptor ACE2, followed by a substantial structural rearrangement to fuse the viral membrane with the host-cell membrane[1–11]. As a viral surface-exposed structural protein, S is targeted by human immune system for production of neutralizing antibodies as a key antiviral mechanism. Indeed, a large number of SARS-CoV-2 neutralizing monoclonal antibodies (MAbs) have been identified, all of which are directed to S, especially its RBD[12–25].

SARS-CoV-2 has been undergoing substantial evolution since its initial emergence[26–32]. Several SARS-CoV-2 lineages have been classified by the World Health Organization (WHO) as variants of concern (VOC), including B.1.1.7 (Alpha), B.1.351 (Beta), P.1 (Gamma), and B.1.617.2 (Delta), each of which harbors multiple mutations in the S protein. These variants rapidly became the dominant strains locally, leading to large second waves of infection, and they continue to spread globally[31]. Among them, the Delta variant emerged in October 2020 in India[29,33], but have spread rapidly to up to 135 countries as of 3 August 2021[34]. This variant has become the predominant strain in many countries[29,35]. It has been documented that the Delta variant is 1.1- to 1.4-fold more transmissible than the Kappa (B.1.617.1) and Alpha variants[36]. The Delta variant exhibits higher replication efficiency in cultured airway organoid and human epithelial systems[36] and in human individuals compared to the original strain[35,37]. Also, the Delta variant is less sensitive to serum neutralizing antibodies from recovered individuals or vaccine recipients, and even compromises the neutralizing potency of some MAbs[30,31,36]. All these characteristics of Delta make this variant particularly threatening. Therefore, further understanding of the nature of Delta is of significant importance and may help in developing countermeasures against this VOC.

Different SARS-CoV-2 variants harbor diverse patterns of mutations in their S protein, which may be associated with improved virus fitness and immune evasion[38,39]. Previous studies have investigated the impact of such mutations on the spike structure, receptor binding, and antigenicity of the Alpha[40–43], Beta[44,45], Gamma[46,47], Epsilon (B.1.427/B.1.429)[48], D614G[26,49], and N501Y[10,41] variants as well as the partially mutated Delta RBDs[31]. The Delta variant harbors multiple mutations in S protein, including two substitutions (L452R and T478K) in the RBD, the D614G substitution, a cluster of mutations in the N-terminal domain (NTD), and two more substitutions near the furin cleavage site and in the heptad repeat 1 (HR1). As a result, Delta can escape the neutralization by some MAbs against the original strain[30,31,36]. In addition, the Delta variant showed higher binding affinity between RBD and ACE2 receptor[31] and 10-fold increased spike-mediated entry efficiency compared to the wild type (WT) strain[36]. However, due to lack of high-resolution structural information especially on the Delta S trimer in complex with ACE2 receptor or neutralizing MAbs, how spike variations impact virus fitness, transmissibility, and neutralization sensitivity remain to be elucidated.

In this study, we present two cryo-EM structures of the Delta S trimer in the open and transition state at 3.1- and 3.4-Å-resolution, respectively. Combined with 3D variability analysis (3DVA), we show that RBD destabilizations, mediated by alternation of RBD-RBD contact, lead to a large population shift towards the more RBD-up and S1 destabilized fusion-prone state, beneficial for engagement with ACE2 and shedding of S1. We further capture four states for the Delta S/ACE2 complex at 3.2- to 3.6-Å-resolution, depicting high propensity for more RBD-up conformation for receiving more ACE2s. Importantly, our structural and biochemical data demonstrate that the Delta T478K substitution plays a vital role in stabilization and reshaping of the loop[473–490] of the receptor binding motif (RBM), leading to enhanced interaction with ACE2. Moreover, we identify the previously generated MAb 8D3[23] as a broadly neutralizing antibody that potently cross-neutralizes Delta, Beta, and Kappa variants. Our structural study reveals that the 8D3 epitope is located in the RBM loop[473–490] region, and 8D3 binding induces a large side-chain orientation alternation of the substituted K478, suggesting 8D3 may use a unique "induced-fit" mechanism unreported for SARS-CoV-2 neutralizing antibodies to tolerate Delta T478K mutation. We also found that all five RBD-targeting neutralizing MAbs tested remain effective on Delta whereas two of them lose potency towards Beta, indicating that Delta better preserves the neutralizing antigenic landscape in the RBD region as compared to Beta.

## Results

**Structural variation and conformational dynamics of the Delta variant S trimer.** To visualize the impact of the substitutions on the spike conformation, we prepared a prefusion-stabilized trimeric S protein of SARS-CoV-2 Delta variant (Supplementary Fig. 1) and determined two cryo-EM structures of the Delta variant S trimer. The two cryo-EM maps, including a one RBD-up open conformation (termed Delta S-open) and a transition state (termed Delta S-transition), were obtained at 3.1- and 3.4-Å-resolution, respectively (Fig. 1a, b; Supplementary Fig. 2a–c, Supplementary Table 1). We then built an atomic model for each of the two structures (Fig. 1c, Supplementary Fig. 2d). There is no linoleic acid (termed LA) in the Delta S-open and S-transition maps, as in our recent Kappa and Beta S structures, obtained in the same construction and purification condition[50]. LA binding has been detected in the tightly closed WT S trimer structures[51–53] and been suggested to lead to more compacted RBDs[51]. The population distribution of the Delta S-open and S-transition is about 75.3% and 24.7% (Fig. 1d), respectively, displaying a considerable population distribution shift to the open state than that of our Kappa and Beta S structures (both around 50–50% open-transition ratio)[50]. The population of our Delta S-open is comparable to that of the recent reported Delta S (~70%)[54], also higher than those of the S trimers in the absence of LA (ranging from ~32% to 50%)[49,55], except for one case with all the WT S in open state[56].

Here, for the S-open state, the up RBD-1 angle is about 71.5° (Fig. 1e, f), 2.4° more open than that (69.1°) of the parental strain D614G (termed G614) S-open (PDB 7KRR)[49]. Besides, the NTDs clockwise rotated up to 7.0° relative to the G614 S-open (Fig. 1g), which is accompanied by noticeable outward tilts of NTD, loop[290–303], and helix[920–940] of HR1 (Fig. 1h), collectively leading to a less compact S-open conformation for Delta. Moreover, in Delta S-open, all three fusion peptides (FP, residues 828 to 853) are disordered, and the 630 loop (residues 620 to 640) in the RBD-up protomer 1 is disordered while those in the RBD-down protomer 2/3 are partially disordered (Fig. 1i). In contrast to this, the 630 loop in the RBD-down protomer 3 of G614 S-open remains structured[45,49]. It has been proposed that the FP and 630 loop modulate the stability of the SARS-CoV-2 S protein and structured ones would help clamp down the RBDs[45,49]. Thus, the more disordered 630 loop in all the three Delta S protomers could reduce constrains on RBDs, resulting in higher propensity for RBD-up state of the Delta spike and improved virus fitness.

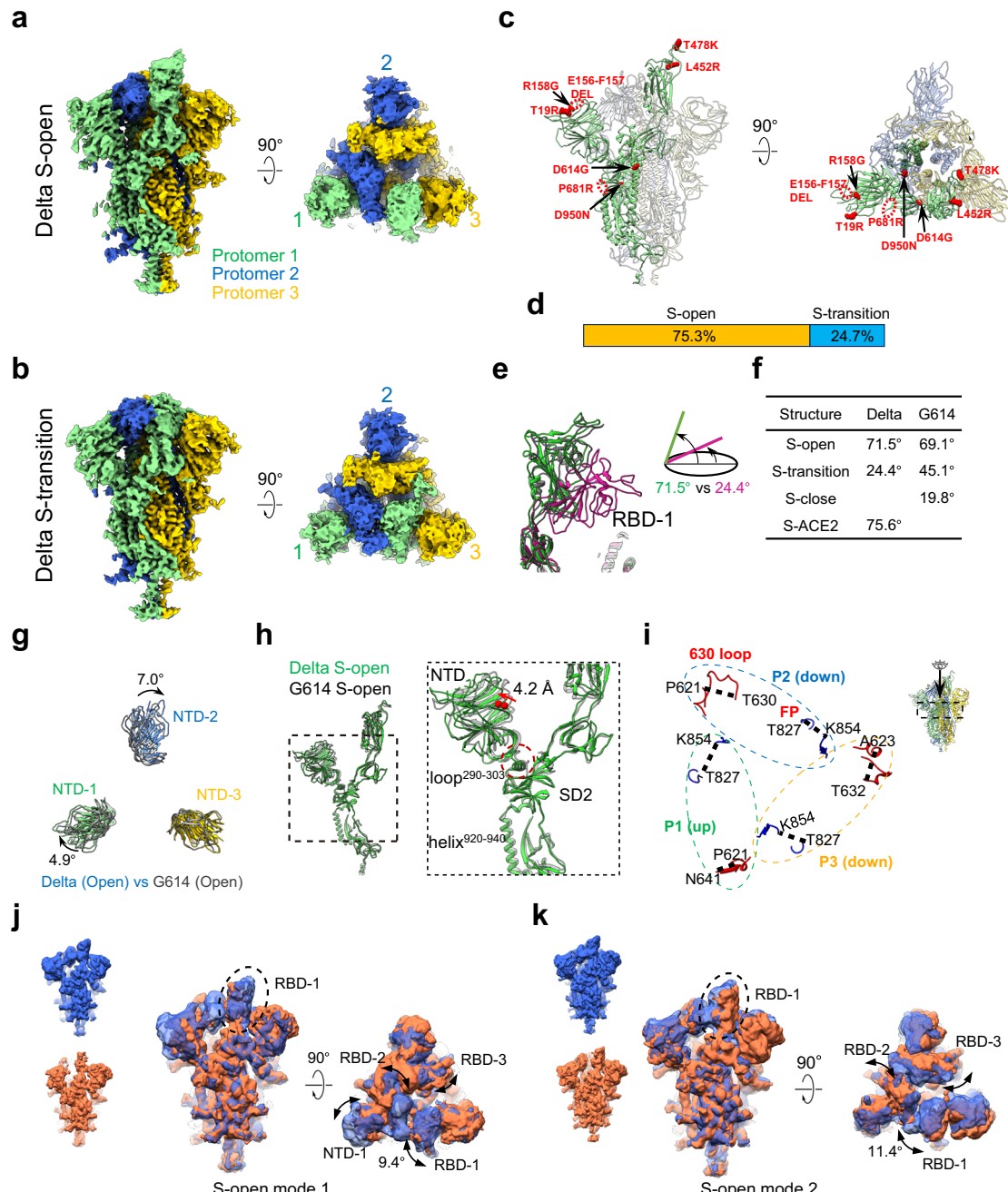

**Fig. 1 Cryo-EM structures of the SARS-CoV-2 Delta variant S trimer. a, b** Cryo-EM maps of the Delta variant S-open **a** and S-transition **b** state. Protomer 1, 2, and 3 are shown in light green, royal blue, and gold, respectively, which color scheme was followed throughout. **c** Atomic model of the Delta S-open, and Delta mutations indicated by red sphere. **d** Population distribution of the Delta S-open and S-transition. **e** Overlaid RBD-1 from Delta S-open (light green) and S-transition (violet red), showing that the angle between the long axis of RBD and the horizontal plane of S trimer is reduced from S-open to S-transition. The RBD-1 of G614 S-open (PDB 7KRR, grey) also showed here as a comparison. **f** List of the angle between RBD-1 and the S-trimer plane in different states of Delta S, compared with that of the G614 variant[45,49]. **g** Top view of the overlaid NTDs of the Delta S-open (in color) and the G614 S-open (PDB 7KRR, gray), indicating a clockwise rotation/untwist of the Delta S relative to that of G614. **h** Side view of the overlaid protomer 1 of S-open, showing an outward tilt of NTD (~4.2 Å), associated with small outward shift of NTD loop[290-303] and HR1 helix[920-940], relative to that of G614 S-open (PDB 7KRR, gray). **i** In Delta S-open, all three FPs (blue) are disordered, the 630 loop in the RBD-up protomer 1 is disordered, and that in the RBD-down protomer 2/3 is partially disordered. **j, k** Two representative 3DVA motions of the Delta S-open dataset. The left two maps illustrate the two extremes in the motion, and the angular range and direction of the motion are displayed in the two views of the overlaid two extreme maps (right).

For the S-transition state, the angle between the long axis of RBD-1 and the horizontal plane of S trimer is 24.4° (Fig. 1e, f), upwards tilted about 4.6° relative to the down RBD in G614 S-closed (PDB 7KRQ)[49], and its NTDs untwisted 7.0° to 8.4° compared with that in the G614 S-closed (Supplementary Fig. 2e).

In addition, in our Delta S-transition structure, the three protomers appear asymmetric (Supplementary Fig. 2g), and the 630 loops and FPs are partially or fully disordered (Supplementary Fig. 2i), which is usually the case in the transition or open state of WT/G641/Delta spike[49,53,54]. Collectively, these features

suggest that the S-transition is not in the closed state, instead, it may represent an open initiation transition state with its RBD-1 in the initial stage of lifting up. Compared to the recent study on the full-length membrane embedded Delta spike[54], our S-open appears more comparable to their S-open-2 state (Supplementary Fig. 2f), and our Delta S-transition is more untwisted/open (up to 6.5°) with the RBD-1 tilted upwards slightly than that of their Delta S-closed state (Supplementary Fig. 2h). Still, we can't rule out the possibility that the results generated from our prefusion stabilized ectodomains may not fully reflect what is seen in the full-length, membrane embedded spike. Moreover, compared to a recent bioRxiv report[57], our S-open exhibits similar confirmation to one of their higher populated one-RBD-up S-open states (7V7Q, Supplementary Fig. 2f).

To examine the intrinsic conformational dynamics of the Delta S trimer, we performed further 3DVA on the Delta S-open dataset through cryoSPARC[58]. This analysis revealed a motion, in which the original contact between RBD-1 and RBD-2 is released with an RBD-1 swing motion towards RBD-2 direction in an angular range of 9.4°, destabilizing both RBD-1 and RBD-2, and RBD-2 moves away with an upward tilt; in the meanwhile, NTD-1 tilts inwards to form contact with RBD-2, and RBD-3 swings accompanying RBD-2 movement (Fig. 1j, Supplementary Movie 1). This indicates RBD destabilizations and especially an intrinsic transient RBD-2 raising up motion mediated by alternation of RBD-RBD contact, beneficial for ACE2 receptor binding. In addition, we also captured a "breath" motion of the fusion machinery initiated by an RBD-1 swing movement up to 11.4°, in which the three inter-protomer NTD-RBD pairs (NTD-1/RBD-2, NTD-2/RBD-3, and NTD-3/RBD-1) tilt outward and downward simultaneously (Fig. 1k, Supplementary Movie 2), destabilizing S1 relative to S2. This motion may release the original inter-protomer constrains, beneficial for the transient raising up of more RBDs and shedding of S1, rendering the Delta variant S prone to receptor binding and subsequent fusion, leading to improved virus fitness.

**Structural basis of enhanced S-ACE2 interaction for the Delta variant.** The Delta variant bears two mutations (T478K and L452R) in the RBD region compared with the WT strain. We evaluated whether the human ACE2 receptor-binding ability of the Delta variant S trimer is affected by performing biolayer interferometry (BLI) assay. The S trimer of the G614 variant, which only carries a D614G mutation relative to the WT strain[26,49], was also analyzed for ACE2 binding for comparison purpose. We found that the Delta S showed higher ACE2-binding affinity (equilibrium dissociation constants [KD] = 41 nM) than the G614 S (KD = 87 nM) (Fig. 2a), in line with other reports[31].

To uncover the structural mechanism underlying the observed binding property changes, we carried out cryo-EM study on the Delta S trimer in complex with human ACE2 peptidase domain (PD) domain (Supplementary Fig. 3). We obtained four cryo-EM maps (Fig. 2b), including a conformation with one RBD up and engaged with an ACE2 (termed Delta S-ACE2-C1), two conformations that contain two "up" RBDs, including ACE2-bound RBD-1 and one of the RBD-2 or RBD-3 is also "up" (termed Delta S-ACE2-C2a and S-ACE2-C2b, respectively), and one conformation in which all three RBDs are up and RBD-1 is stably bound with ACE2 (termed Delta S-ACE2-C3), at 3.6-, 3.4-, 3.4-, and 3.2-Å-resolution, respectively (Supplementary Fig. 4a, c, and Supplementary Table 1). In the C2a/C2b/C3 maps, other than the stably associated ACE2 in RBD-1, densities of other associated ACE2s appear weaker. We then built an atomic model for each of the four structures (Supplementary Fig. 4d). The Delta S-ACE2-C1 structure revealed that engagement with ACE2

induces a further 4.1° upward tilt (from 71.5° to 75.6°) of RBD-1 relative to the counterpart in Delta S-open (Fig. 1f). Noteworthy, compared with the Beta and Kappa S-ACE2 complex from our recent study[50] (the three-RBD-up C3 state at 27.7% and 34.1% population, respectively, purified and assembled in the same condition for the three systems), the Delta S-ACE2 complex showed a considerable population shift towards the more ACE2 engaged C3 state (46.6%, Fig. 2c), which could be beneficial for subsequent S1 shedding and the S trimer transition towards postfusion state.

We further focus-refined the stably associated Delta RBD-1-ACE2 region to 3.4-Å-resolution (Fig. 2d and Supplementary Fig. 4b), which clearly depicts the side-chain densities of the mutated RBM L452R and T478K (Fig. 2e). Inspection of the RBD-1-ACE2 interaction interface revealed that the RBM loop[473−490], which plays important roles in the interactions with ACE2 receptor and neutralizing MAbs[8,23,59], exhibits observable conformational change induced by T478K substitution (Fig. 2f, indicated by red arrow). Specifically, the substituted K478 forms a new hydrogen bond (H-bond) with the main chain of N487 (also resides in loop[473−490]), potentially stabilizing the key RBM loop[473−490] (Fig. 2f and Supplementary Table 2, 3). Of note, this substitution also results in the formation of two new H-bonds with ACE2 (including Y489 with ACE2 Y83, and F490 with ACE2 K31, Fig. 2f and Supplementary Table 2), enhancing the S-ACE2 interaction. Further inspection of the surface property showed that the T478K substitution makes the substituted site more positively charged and hydrophilic, which may strengthen RBM interaction with the negatively charged and hydrophilic ACE2 in the interaction interface (Fig. 2g–i). Corroborating this, the Delta variant RBD-ACE2 interaction area (928.4 Å²) was enlarged compared to that of the WT (843.3 Å²) (Fig. 2j and Supplementary Fig. 4e). Collectively, this Delta variant RBM T478K substitution could stabilize and induce conformational change of the RBM loop[473−490] and strengthen the interaction with ACE2 receptor (Fig. 2a), potentially leading to enhanced transmissibility of the Delta variant[36]. Another mutation in the Delta variant RBD, L452R, is not involved in the interaction with ACE2 (Fig. 2e), however, this mutation results in a hydrophilic, basic local surface property reverse, and an elongated side chain (Fig. 2g, h), which may affect the binding of MAbs targeting this region, potentially leading to immune evasion.

**Conformational dynamics of the ACE2 associated Delta S trimer.** We further examined the conformational dynamics of the Delta S-ACE2 complex through 3DVA. Here, mode 1 displays an obvious swing motion of the associated RBD-1-ACE2 approaching/leaving RBD-2 in an angular range of 9.7° with the associated NTD-3 (Fig. 3a, Supplementary Movie 3), in a direction also seen in the WT S-ACE2 complex[8]. This motion represents conformational dynamics of the S-ACE2-C2a state, which to some extend disturbs the RBD-2/RBD-3 interaction. In mode 2, a noticeable outward movement of RBD-1-ACE2 leads to an upwards tilt of RBD-2 to the up position, associated with a downward/outward tilt of the contacting NTD-1; in the meanwhile, RBD-3 moves towards RBD-2, disturbing the original contact with the neighboring NTD-2 (Fig. 3b, Supplementary Movie 4). This motion depicts the transformation from the S-ACE2-C1 to C2a state with RBD-2 gradually opened. Moreover, mode 3 shows that with an ACE2 binding induced swing motion of RBD-1-ACE2 in an angular range of 12.2° along the NTD-1 to NTD-3 direction, the original contact between RBD-2 and RBD-3 is disturbed, leading to simultaneous upwards tilting of RBD-2 and RBD-3 (Fig. 3c, Supplementary Movie 5). This motion reveals the transition from S-ACE2-C2a to the all-RBD-up

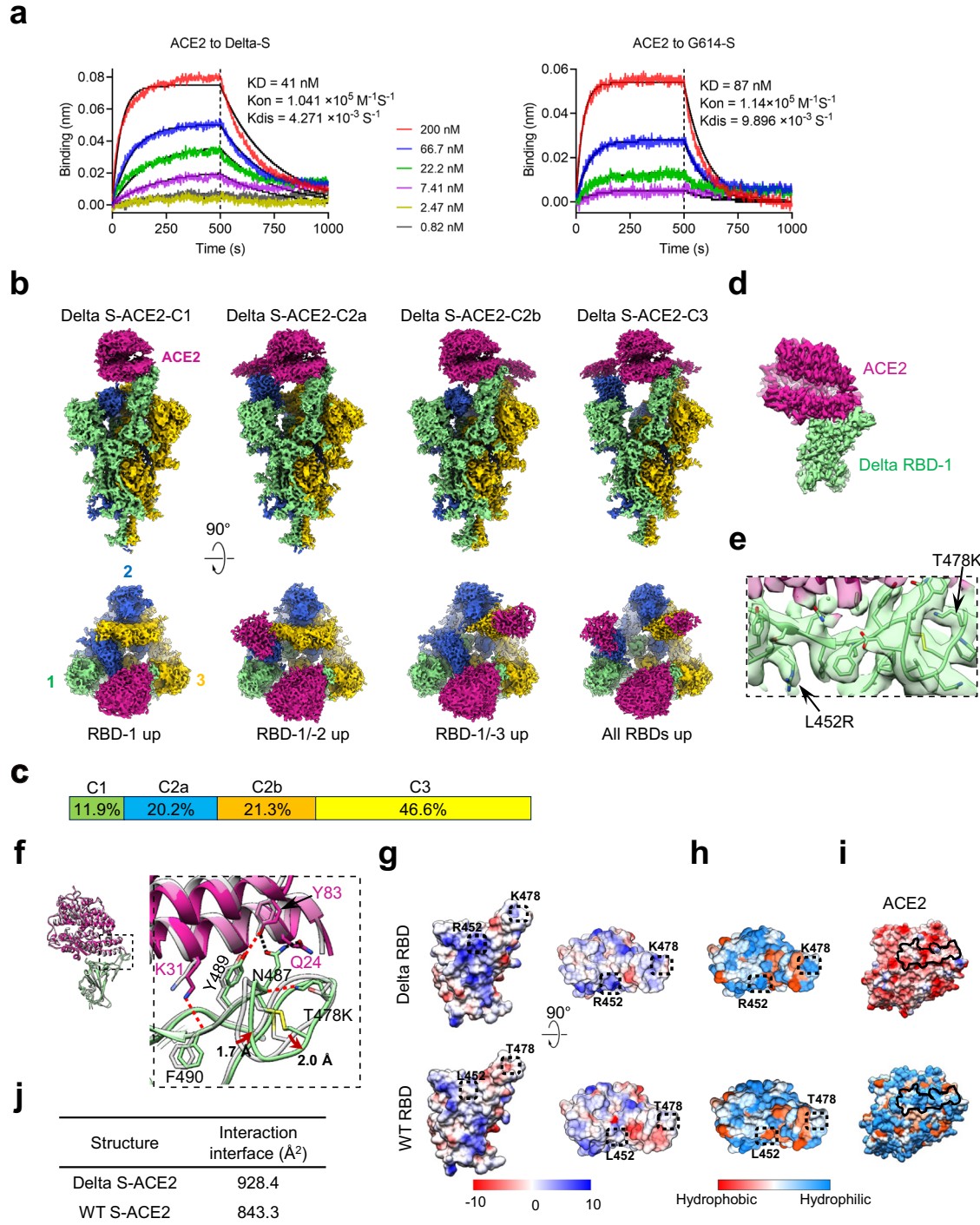

**Fig. 2 Structural basis of enhanced Delta variant S trimer/ACE2 interaction. a** Measurement of the binding affinity between ACE2 and the S trimer of the Delta (left) or G614 (right) variants using bio-layer interferometry (BLI). Association and dissociation steps were divided by dotted lines. ACE2 concentrations tested were shown. Raw sensor grams and fitting curves were shown in color and black, respectively. Source data are provided as a Source Data file. **b** Cryo-EM maps of the Delta S-ACE2 complex in four distinct conformational states. ACE2 is shown in violet red. This color scheme is followed throughout. **c** Population distribution of the Delta S-ACE2 conformers. **d** Density map of the focus-refined Delta RBD-1-ACE2. **e** Zoomed-in view of the S-ACE2 interaction interface, showing the side chain densities of the substituted L452R and T478K were well resolved. **f** The substituted K478 forms a new H-bond with N487, resulting in the formation of two new H-bonds with ACE2 (red dotted line), and a conformational change of loop[473–490] (indicated by red arrow) relative to that in WT RBD-ACE2 (PDB 6M0J). **g, h** The electrostatic surface property **g** and hydrophilicity and hydrophobicity properties **h** of the Delta and WT RBDs, with the mutated residues indicated. **i** Similar surface properties for ACE2, with residues in proximity to Delta RBD-1 (< 4 Å) indicated. **j** Interaction interface areas between ACE2 and RBD of Delta or WT (PDB 6M0J) strain analyzed using PISA.

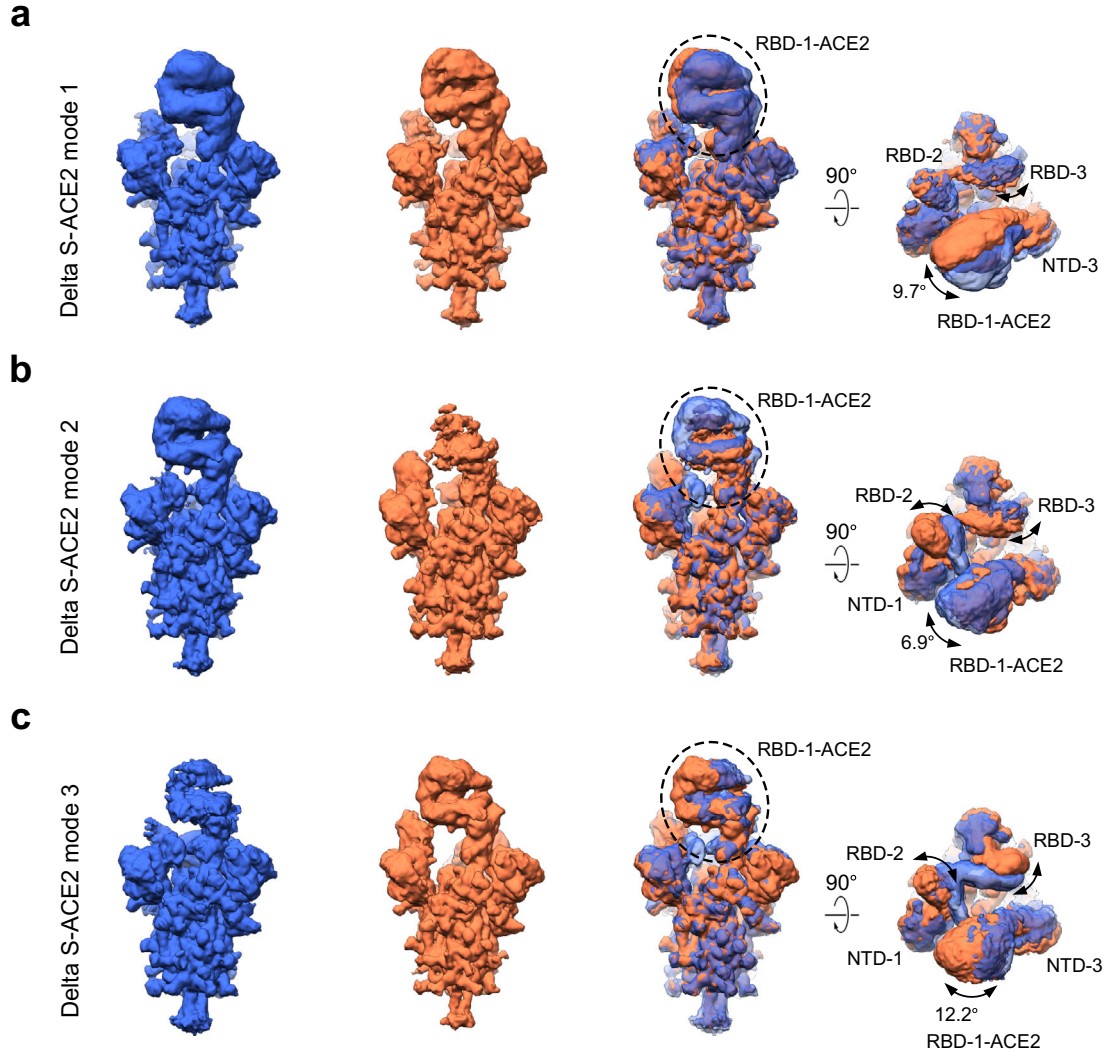

**Fig. 3 3D variability analysis on the delta S-ACE2 complex. a–c** Three representative 3DVA modes of the Delta S-ACE2 complex, with the left two maps showing the two extremes in the motion, and the angular range and direction of the motion displayed in the two views of the overlaid two extreme maps (right).

C3 state. Taken together, our 3DVA on Delta S-ACE2 complex captured how ACE2-binding induced RBD-1 movements destabilize key structural elements and eventually S1 as a whole, revealing that such destabilization is achieved through disturbing RBD-RBD interactions and lead to conformational transitions to more RBD-up states.

**MAb 8D3 is a cross-variant broadly neutralizing antibody**. To assess the impact of the Delta variant on antibody neutralization, we tested five neutralizing MAbs isolated in our previous study for neutralization of a panel of pseudoviruses representing major SARS-CoV-2 variants, including Delta, Beta, and Kappa. These five MAbs, including 2H2, 2G3, 3C1, 3A2, and 8D3, were originally raised against the RBD of the SARS-CoV-2 WT strain[23]. As shown in Fig. 4a, b, MAbs 2H2 and 3A2 remained neutralizing against Delta but were ineffective on Beta and Kappa; MAb 2G3 could efficiently neutralize all three variants despite its potency against Delta reduced by 12 folds compared to that against the WT strain; MAb 3C1 was able to cross neutralize all three variants with relatively low efficiency (IC50s ranging from 338 to 1460 ng/mL); notably, MAb 8D3 exhibited potent cross-neutralization with IC50s against Delta, Beta, and Kappa being 7.6, 6.3, and 2.2 ng/mL, respectively, comparable to those against

WT pseudovirus (IC50 = 7 ng/mL)[23]. These data show that 8D3 and 3C1 are cross-variant broadly neutralizing MAbs and suggest that most, if not all, of the neutralizing epitopes in the RBD region are maintained on the Delta variant S protein. In contrast, mutations in the Beta and Kappa S proteins, especially the E484K or E484Q substitution, may greatly impact RBM-targeting neutralizing MAbs.

We then evaluated the binding ability of the five MAbs to the WT and Delta S proteins by ELISA. As shown in Fig. 4c, for MAbs 2H2, 3C1, 3A2, and 8D3, their reactivity profile with the Delta S closely resembled that towards the WT S, whereas 2G3 exhibited decreased binding of the Delta S relative to the WT. We further analyzed these five MAbs for binding three recombinant RBD proteins derived from the WT strain, the Delta variant, and the Beta variant. As shown in Fig. 4d, binding profiles of the MAbs with WT-RBD or Delta-RBD were similar to those with the WT S or Delta S (Fig. 4c), while 2H2 and 3A2 showed no reactivity to the Beta-RBD. Overall, the antigen-binding ability of the MAbs was in good agreement with their neutralization potency towards specific variant pseudovirus (Fig. 4a, b).

Collectively, the above results demonstrate that 8D3 and 3C1 antibodies possess cross-variant binding and neutralization abilities. The observed broad neutralization by MAb 3C1 is as

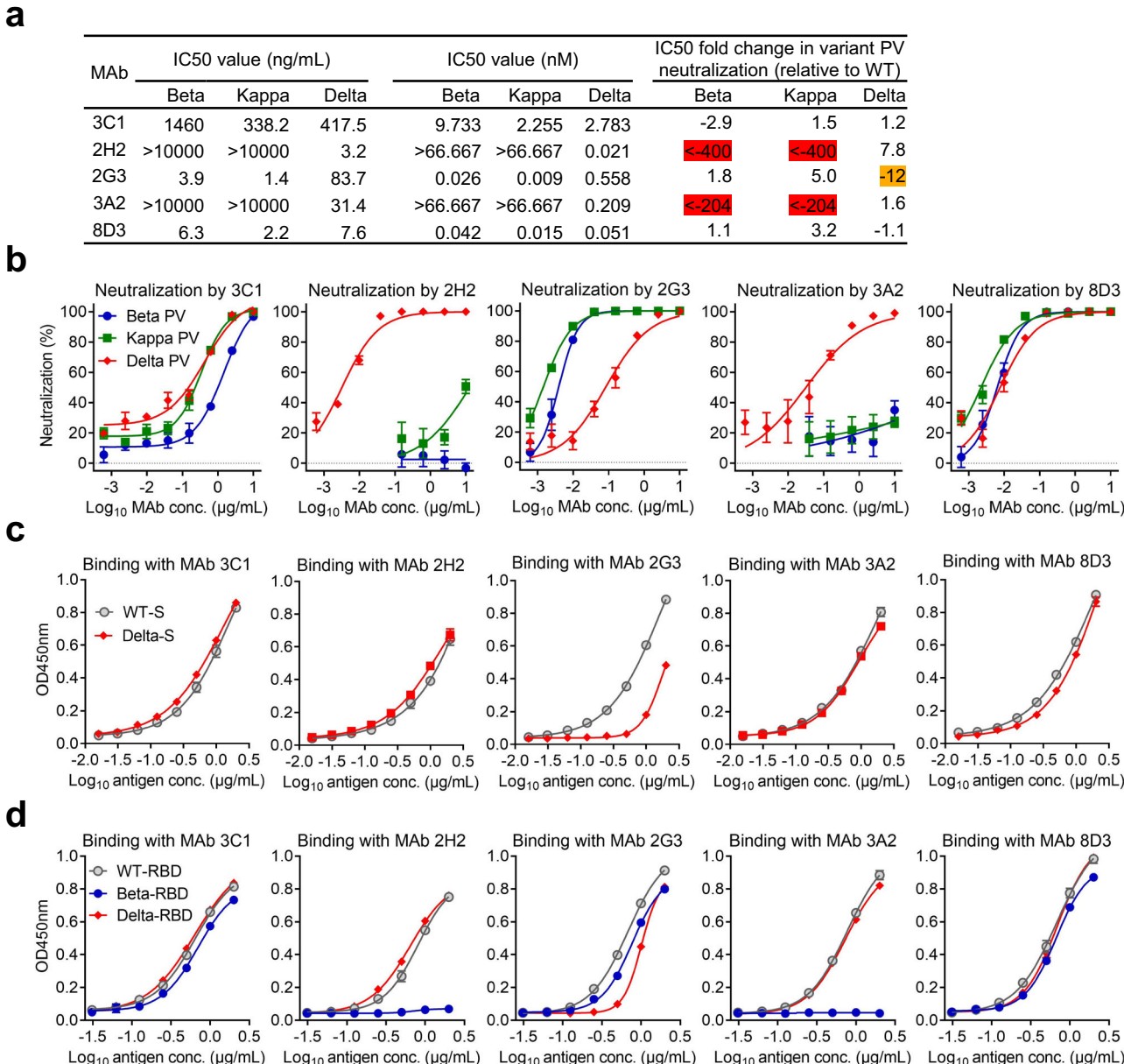

**Fig. 4 Neutralization breadth and binding properties of the MAbs (3C1, 2H2, 2G3, 3A2, and 8D3) against SARS-CoV-2 variants. a** Neutralization values and fold change in neutralization IC50 of the MAbs against the variant pseudoviruses, relative to the WT pseudovirus. A minus sign (-) denotes decrease. Orange shade, more than 10-fold decrease; red shade, more than 100-fold decrease. **b** Neutralization activity of the MAbs towards SARS-CoV-2 Beta, Kappa, and Delta variant pseudoviruses. The MAbs were serially diluted and tested for neutralization. Data are expressed as mean ± SEM of four replicate wells. **c** Binding properties of the MAbs with recombinant S trimers derived from the WT and Delta strains were determined by ELISA. Serial dilutions of S trimers were coated onto the wells. Data are expressed as mean ± SD of triplicate wells. **d** Binding properties of the MAbs with recombinant RBD proteins derived from the WT, Beta, and Delta strains were measured by ELISA. Serial dilutions of RBD were coated onto the wells. Data are expressed as mean ± SD of triplicate wells. Source data are provided as a Source Data file.

expected, because it targets the highly conserved core region of RBD[23]. It is surprising that 8D3 also showed broad binding and neutralization potency towards the variants, despite its epitope being roughly located in the frequently mutating RBM region[23]. In fact, the RBM-targeting MAb 2H2 was greatly affected by the RBM mutations (K417N, E484K, and N501Y) in the Beta variant (Fig. 4a, b, d).

**Structural basis of the broadly neutralizing antibody 8D3 Fab.** To reveal the structural basis of 8D3-mediated broad neutralization of SARS-CoV-2 variants, we carried out cryo-EM study on the Delta S trimer in complex with the Fab fragment of

8D3 and obtained a cryo-EM map at 3.1-Å-resolution (termed Delta S-8D3, Fig. 5a, Supplementary Fig. 5, and Supplementary Table 1). The Delta S-8D3 structure displayed a configuration with RBD-1 in the "up" position and engaged with an 8D3 Fab, while the other two RBDs in the "down" position without associated Fab (Fig. 5a). To improve structural details in the RBD-8D3 interaction interface, we further focus-refined the relatively dynamic RBD-1-8D3 Fab region and obtained a map of this region with enhanced local resolution (Fig. 5b, Supplementary Fig. 5c), revealing side chain densities in the interaction interface and the substituted L452R and T478K (Fig. 5c). The 8D3 Fab binds on the tip of RBD-1 (Fig. 5b), with the CDRs of 8D3

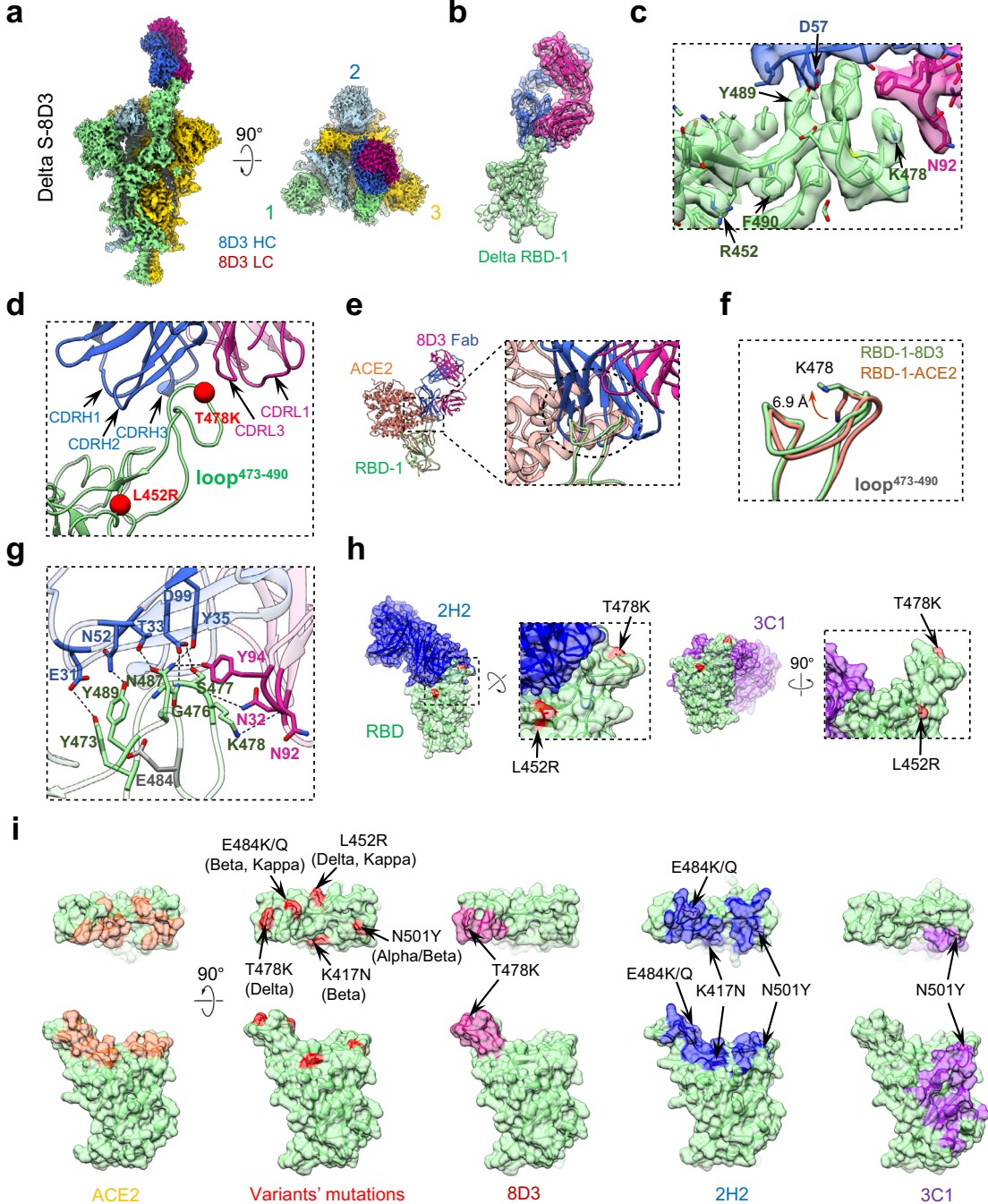

**Fig. 5 Cryo-EM analyses on the Delta S-8D3 Fab complex. a** Side and top views of the cryo-EM map of the Delta S-8D3 complex, with the heavy and light chain of 8D3 Fab in royal blue and violet red, respectively. The color scheme was followed. **b, c** Model-map fitting of the focus-refined Delta RBD-1-8D3 structure **b**, and the zoomed-in view of the Delta RBD-1/8D3 interaction interface **c**. The sidechains of the key amino acids, including R452 and K478, are well resolved. **d** The RBD-1/8D3 interaction interface, with major involved structural elements and the two RBM mutations of Delta variant labeled. **e** ACE2 (coral, from our Delta RBD-1-ACE2 structure) would clash with the heavy chain of 8D3 Fab (dotted black circle). They share overlapping epitopes on the RBM loop[473–490]. **f** The side chain orientation alternation of K478 from Delta RBD-1-8D3 (in light green) relative to that from Delta RBD-1-ACE2 (coral). **g** Magnified view to show the H-bond interaction network formed between 8D3 Fab and the RBM loop[473–490] of Delta variant. **h** Distinct binding of 2H2 (PDB 7DK4, blue) and 3C1 (PDB 7DCC, purple) Fab on the RBD (light green) of WT S. The Delta variant mutation sites (in red surface) are not located in the binding footprint of 2H2 and 3C1 on RBD. **i** The footprint (<4 Å contact) of ACE2 (coral, our Delta RBD-1-ACE2 structure), 8D3 (violet red), 2H2 (blue), and 3C1 (purple) Fab, respectively, on RBD. The second column shows the RBD with mutation sites of the Delta, Beta, and Kappa variants indicated in red.

forming a pocket wrapping around the RBM loop$^{473-490}$ region including the substituted K478, while the other substituted R452 is not involved in the interaction (Fig. 5d, Supplementary Tables 4, 5). Since the loop$^{473-490}$ is also a part of the ACE2 binding sites on RBD, the association of 8D3 Fab to this region could therefore competitively block ACE2 receptor binding with RBD (Fig. 5e).

Moreover, our Delta RBD-1-8D3 structure revealed that the loop$^{473-490}$ exhibits a conformational variation relative to that in the RBD-1-ACE2 structure (Fig. 5f). Particularly, an obvious side chain orientation alternation (~ 6.9 Å) of the substituted K478 was observed. It appears that all the CDRs, except CDRL2, of the 8D3 Fab are involved in the interaction with the loop$^{473-490}$ (Fig. 5d, Supplementary Tables 4, 5). Specifically, all the three CDRHs participate in an extensive interaction with the RBM loop$^{473-490}$ by forming six H-bonds, and CDRL1 and CDRL3 also involve in the interaction with this loop by forming three H-bonds (Fig. 5g, Supplementary Table 4). Here, the N92 of CDRL3 forms a H-bond with the substituted K478 from loop$^{473-490}$ (Fig. 5g), leading to the K478 side-chain orientation change, which breaks the original K478-N487 H-bond within the loop$^{473-490}$, resulting in an observable loop$^{473-490}$ conformational change in Delta RBD-1-8D3 (Fig. 5f).

SARS-CoV-2 variants other than Delta harbor up to three distinct mutations in their RBD regions, e.g., Alpha (N501Y), Beta (K417N, E484K, and N501Y), and Kappa (L452R and E484Q). According to our Delta RBD-1-8D3 Fab structure, N501, K417, and L452 are not located in the interaction interface between 8D3 Fab and RBD (Fig. 5i); while E484, although residing within the loop$^{473-490}$, remains distant from forming contact with the 8D3 Fab (Fig. 5g), thus explaining the observed potent neutralization of these variants by 8D3 (Fig. 4a, b).

As for 2H2 and 3C1 MAbs, the Delta L452R and T478K substitutions are not located in their neutralizing epitopes on RBD (Fig. 5h, i), explaining why 2H2 and 3C1 remain effective against Delta variant. Moreover, for 2H2, the other mutations, including K417N, E484K/Q, and N501Y mostly from the Beta and Kappa variants (N501Y also presented in the Alpha variant), all reside in its neutralizing epitopes on RBD (Fig. 5i), which collectively could affect the interaction between MAb 2H2 and RBD. Indeed, MAb 2H2 failed to neutralize Beta and Kappa variants (Fig. 4a, b). While for the 3C1, only the N501Y substitution from Beta (also Alpha) variant resides in the edge of its epitope on the RBD (Fig. 5i), and considering 3C1 could adopt varied orientations to associate with RBD[23], 3C1 is likely not or minimally affected by the N501Y mutation and therefore remains effective against Beta and Kappa variants in the neutralization experiment (Fig. 4a, b).

## Discussion

The SARS-CoV-2 Delta variant, initially emerged in October 2020 in India[29,33], has spread over one hundred countries and is currently the predominant circulating strain in the world[34,35]. Compared to the original virus, the Delta variant shows higher infectivity and increased resistance to antibody neutralization[30,31,36]. Thus, it is essential to understand how the spike variations of the Delta variant affect virus transmissibility and neutralization sensitivity. In the current study, we performed cryo-EM study and biochemical analysis on the Delta S trimer and its complex with ACE2 receptor. We captured two conformational states for the Delta S trimer, including S-open and S-transition, in the population distribution of 75.3% and 24.7%, respectively (Fig. 1a, b, d). This suggests a considerable population shift towards the open state than that of the Kappa and Beta S trimer from our recent study[50] and other S trimers in the

absence of LA (the open ranging from ~32% to 50%)[49,55], demonstrating a population shift of the Delta S towards the fusion-prone open state, beneficial for ACE2 binding. Moreover, the Delta S-open structure displayed a further untwisted/open conformation with more distorted 630 loops compared with the G614 S-open[45,49], which could reduce constrains on RBDs, resulting in higher propensity for RBD-up state of the Delta spike. In line with this, our 3DVA depicted an intrinsic RBD-2 raising up motion and a "breathing" motion of Delta S especially in the S1 region (Fig. 1j, k, Supplementary Movies 1, 2). These motions revealed how the ACE2-binding induced RBD-1 movements destabilize the key structural elements and eventually S1 as a whole (Fig. 3, Supplementary Movies 3–5), indicating RBD destabilizations mediated by alternation of RBD-RBD contact, is beneficial for raising up of more RBDs and shedding of S1. Corroborating this, the four states we captured for the Delta S-ACE2 complex displayed a considerable population shift towards the more ACE2 engaged C3 configuration (46.6% three RBD-up C3 state, Fig. 2c) than that of the Beta or Kappa S-ACE2 complexes from our recent study (C3 population at 27.7% and 34.1%, respectively)[50]. Collectively, our study revealed structural features of Delta S that make it more efficient in receptor binding and subsequent fusion, providing a possible explanation to the improved virus fitness and higher transmissibility of the Delta variant[36].

Noteworthy, our Delta RBD-1-ACE2 structure showed that the T478K substitution in Delta can stabilize and reshape the RBM loop$^{473-490}$ by forming a new H-bond with N487 also within this loop (Fig. 2f), which induces the formation of two additional H-bonds between loop$^{473-490}$ and ACE2, thus enhancing the binding affinity of Delta S with ACE2 (Fig. 2a, g). In addition, with the T478K substitution, the RBM interaction interface tends to be more positively charged and hydrophilic, beneficial for RBM interaction with the negatively charged ACE2 in related interaction interface (Fig. 2g–i). Taken together, our study demonstrates that the increased binding to ACE2, mediated by both affinity-enhancing RBM T478K substitution and increased propensity for the receptor-accessible RBD-up states, may contribute to the high infectivity of the Delta variant.

SARS-CoV-2 variants gain series of mutations in their S proteins, some of them occurring in the RBD region. As a consequence, VOCs significantly impact the potency of neutralizing antibodies originally developed against WT strains[38,39]. It is thus important to determine whether broadly neutralizing antibodies exist and if yes where they target. In the present study, we screened a panel of five previously isolated RBD-directed neutralizing MAbs[23] for neutralization of Delta, Beta, and Kappa variant pseudoviruses and identified MAb 8D3 as a potent cross-variant neutralizing antibody (Fig. 4). Our further structural study on the Delta S-8D3 Fab complex revealed that 8D3 binds the RBM loop$^{473-490}$ region with its CDRs forming a "crater" to surround the loop$^{473-490}$. The 8D3 binding footprint is relatively small (~623.6 Å$^2$). The residues N501, K417, L452, and E484, where mutations frequently occur in VOCs, are either outside the interaction interface or not directly involved in forming contacts (Fig. 5i), explaining the neutralization potency of 8D3 towards Beta, Kappa and perhaps Alpha and Gamma (P.1) as well. The Delta variant contains a T478K substitution located near the center of the 8D3 binding footprint (Fig. 5d), but surprisingly 8D3 can still efficiently bind the Delta S trimer and RBD and neutralize the Delta pseudovirus (Fig. 4). Our structural study showed that, upon 8D3 binding, the substituted K478 displays a large side-chain orientation change to break the K478-N487 H-bond within the RBM loop$^{473-490}$ but facilitate the formation of a H-bond between K478 and the N92 of CDRL3 (Fig. 5f, g), suggesting that 8D3 may use an "induced-fit" mechanism

unreported previously for SARS-CoV-2 neutralizing antibodies to accommodate the T478K substitution in Delta.

The present study shows that all the five MAbs tested remained neutralizing towards Delta despite the potency of 2G3 reduced by 12 folds relative to that against the WT strain, whereas two of them (2H2 and 3A2) almost completely lost binding and neutralization of Beta (Fig. 4), indicating that neutralizing MAbs raised against the original RBD are less affected by Delta than by Beta. In consistence with our finding, several recent studies reveal that Beta confers higher degrees of resistance to vaccine or convalescent serum neutralization than does Delta and more RBD-targeting neutralizing MAbs are ablated by Beta relative to Delta[30,31,36]. In addition, Delta poses greater impact on NTD-directed MAbs than on RBD MAbs[30,31,36]. Taken together, the results from this and the above-mentioned studies suggest that, compared to Beta, Delta better preserves most if not all of the neutralizing antigenic sites of the WT RBD to allow efficient binding and neutralization by potent antibodies generated against the original strains, inferring that the currently used vaccines (developed based on the original strains), especially those using RBD as the antigen (e.g., ZF2001)[60], may still be efficacious against Delta.

In summary, we determined cryo-EM structures of the Delta S trimer and its complexes with ACE2 receptor. These structures, in combination with 3DVA and biochemical analysis, reveal that the Delta S gains increased binding to ACE2, mediated by both increased propensity for the more RBD-up states and affinity enhancing RBM T478K substitution, thus providing a structural explanation to the enhanced transmissibility and rapid spread of the Delta variant. Moreover, we identify an unusual cross-variant neutralizing MAb 8D3 that targets the RBD and elucidate the structural mechanism for 8D3-mediated broad neutralization. Our structural study also reveals that Delta well preserves original neutralizing epitopes on its RBD and hence remains highly sensitive to most of RBD-directed MAbs. Our findings shed new lights on the pathogenicity and antibody neutralization of the Delta variants, providing important information for controlling this threatening pathogen.

## Method

**Expression and purification of recombinant proteins**. Wild type SARS-CoV-2 S protein was prepared as the published protocol[8]. Briefly, the prefusion-stabilized S ectodomain (residues M1–Q1208) of SARS-CoV-2 Wuhan-Hu-1 strain (GenBank ID: MN908947.3) was produced in the transfected HEK293F cells and then purified from the culture supernatant using Ni-NTA affinity resin. To prepare the constructs of prefusion-stabilized S proteins of SARS-CoV-2 G614 and Delta (B.1.617.2) variants, D614G amino acid substitution of G614 variant and the mutations of Delta variant (T19R, E156DEL, F157DEL, R158G, L452R, T478K, D614G, P681R and D950N) were introduced by site-directed mutagenesis, using our previous prefusion-stabilized SARS-CoV-2 S-trimer expression plasmid[8]. SARS-CoV-2 variants S proteins, and human ACE2 were prepared according to the published protocol[8]. Briefly, the constructs were transiently transfected into HEK293F cells using polyethylenimine (PEI). Three days after transfection, the supernatants were harvested by centrifugation, and then passed through 0.45 μm filter membrane. The clarified supernatants were added with 20 mM Tris-HCl pH 7.5, 200 mM NaCl, 20 mM imidazole, 4 mM MgCl₂, and incubated with Ni-NTA resin at 4 °C for 1 h. The Ni-NTA resin was recovered and washed with 20 mM Tris-HCl pH 7.5, 200 mM NaCl, 20 mM imidazole. The protein was eluted by 20 mM Tris-HCl pH 7.5, 200 mM NaCl, 250 mM imidazole. Similarly, recombinant RBD derived from SARS-CoV-2 Beta or Delta variants were generated using the HEK 293 F expression system. Specifically, recombinant plasmids coding Beta-RBD (the K417N, E484K, and N501Y mutations) or Delta-RBD (the L452R and T478K mutations) were made based on the plasmid pcDNA3.4-SARS-2-RBD[23] by using the Mut Express™ II Fast Mutagenesis Kit V2 (Vazyme, China). The resulting mutant plasmids were separately transfected into HEK293F cells using PEI and cultured for 5 days. The culture supernatants were then collected by centrifugation and loaded onto the pre-equilibrated Ni-NTA affinity column. After wash, the bound His-tagged mutant RBD proteins were eluted with elution buffer (20 mM Tris-HCl pH 7.9, 500 mM NaCl, 250 mM imidazole). The eluted proteins were analyzed by SDS-PAGE and dialyzed against PBS, and protein concentration was determined by Bradford method.

**Bio-layer interferometry (BLI) assay**. Prior to BLI assay, purified recombinant S trimer proteins of the Delta and G614 SARS-CoV-2 variants were first subjected to gel filtration chromatography using a Superose 6 increase 10/300 GL column (GE Healthcare) pre-equilibrated with 20 mM Hepes pH 7.5, 200 mM NaCl, and then biotinylated using the EZ-Link™ Sulfo-NHS-LC-LC-Biotin kit (Thermo Fisher), and Zeba™ spin desalting columns (Thermo Fisher) were then used to remove excess biotin. Binding affinities of S trimers to ACE2 were tested on an Octet Red96 instrument (Pall FortéBio, USA) according to manufacturer's protocol. Briefly, biotinylated S trimers were loaded onto streptavidin (SA) biosensors (Pall FortéBio) until saturation. The S-immobilized biosensors were dipped into wells containing different concentrations of ACE2 monomer protein and then incubated for 500 s. Next, the biosensors were dipped into dissociation buffer (0.01 M PBS with 0.02% Tween 20 and 0.1% bovine serum albumin) and incubated for 500 s. The data were corrected by subtracting reference sample and then fitted to a 1:1 binding model for determination of affinity constants using the software Octet Data Analysis 11.0.

**Pseudovirus neutralization assay**. Murine leukemia virus (MLV)-based pseudoviruses bearing the S protein of SARS-CoV-2 variants were produced following our previously reported protocol[23]. Briefly, the plasmids encoding the full-length S proteins derived from wild-type, Beta, Kappa, or Delta SARS-CoV-2 strains were made and used to produce the corresponding pseudoviruses.

Pseudovirus neutralization assay was carried out with human ACE2-overexpressing HEK 293 T cells (293T-hACE2) following our previously reported protocol[23]. Two days post pseudovirus infection, luciferase activity was determined, and percent neutralization was calculated. For each antibody, half inhibitory concentration (IC50) was calculated using GraphPad Prism software (version 8).

**ELISA**. To determine binding activities of the variants-derived S or RBD antigens with anti-SARS-CoV-2 MAbs developed in our previous study[23], serial dilutions of recombinant S trimer or RBD proteins derived from WT[23], Beta, or Delta SARS-CoV-2 strains were coated onto ELISA plates at 4 °C overnight. After blocking with 5% milk in PBS-Tween 20 (PBST), the plates were incubated with 50 ng/well (1 μg/mL) of the MAbs 3C1, 2H2, 2G3, 3A2, or 8D3[23] at 37 °C for 2 h, followed by incubation with horseradish peroxidase (HRP)-conjugated anti-mouse IgG (Sigma, 1/10,000 dilution) for 1 h at 37 °C. After washes and color development, absorbance was determined at 450 nm.

**Delta variant S trimer/8D3 Fab complex formation**. The Delta variant S trimer/8D3 Fab complex was prepared as described in previous report[23]. Briefly, purified 8D3 IgG was incubated with papain (300:1 W/W) in PBS buffer (in the presence of 20 mM L-cysteine and 1 mM EDTA) for 3 h at 37 °C. The reaction was quenched by 20 mM iodoacetamide. Fab was purified by running over a HiTrap DEAE FF column (GE Healthcare) pre-equilibrated with PBS. Delta S protein was incubated with 8D3 Fab in a 1:8 molar ratio on ice for 1 h. The Delta S-8D3 Fab complex was purified by size-exclusion chromatography using Superose 6 increase 10/300 GL column (GE Healthcare) in 20 mM Tris-HCl pH 7.5, 200 mM NaCl, 4% glycerol. The complex peak fractions were concentrated and assessed by SDS-PAGE.

**Cryo-EM sample preparation**. To prepare the cryo-EM sample of the Delta S trimer, a 2.2 μl aliquot of the sample (~3 mg/ml) was applied on a plasma-cleaned holey carbon grid (R 2/1, Cu, 200 mesh; Quantifoil). The grid was blotted with Vitrobot Mark IV (Thermo Fisher Scientific) using a blot force of -1 and 1 s blotting time at 100% humidity and 8 °C, and then plunged into liquid ethane cooled by liquid nitrogen. To prepare the cryo-EM sample of the Delta S-ACE2 complex, purified Delta S trimer was incubated in a 1:4 molar ratio with ACE2 on ice for 20 min and then vitrified using the same condition described above except that using different grid (R 1.2/1.3, Cu, 200 mesh; Quantifoil). The purified Delta S-8D3 complex was vitrified using the same procedure as for the Delta S-ACE2 sample.

**Cryo-EM data collection**. Cryo-EM movies of the samples were collected on a Titan Krios electron microscope (Thermo Fisher Scientific) operated at an accelerating voltage of 300 kV. For the Delta S trimer, the movies were collected in a magnification of 81,000× and recorded on a K3 direct electron detector (Gatan) operated in the counting mode (yielding a pixel size of 0.893 Å) under a low-dose condition in an automatic manner using EPU software (Thermo Fisher Scientific). Each frame was exposed for 0.05 s, and the total accumulation time was 2 s, leading to a total accumulated dose of 50.2 e⁻/Å² on the specimen. For the Delta S-ACE2 or S-8D3 complex, movies were collected with a magnification of 64,000× (yielding a pixel size of 1.093 Å). Each frame was exposed for 0.1 s, and the total accumulation time was 3 s, leading to a total accumulated dose of 50.2 e⁻/Å² on the specimen.

**Cryo-EM 3D reconstruction**. For each dataset, the motion correction of image stack was performed using the embedded module of Motioncor2 in Relion 3.1[8,61,62] and CTF parameters were determined using CTFFIND4[63] before further data processing. Unless otherwise described, the data processing was performed in

Relion3.1. For the Delta S dataset (Supplementary Fig. 2), we obtained 970,556 particles by automatic particle picking and 506,346 particles remained after reference-free 2D classification. The cleaned-up particles were used for further reconstruction with the WT S-open map (EMD-21457) as initial model[55]. After 3D classification and focused 3D classification on the RBD-1, we obtained a Delta S-open map from 103,096 particles and a S-transition map from 33,761 particles. After Bayesian polishing and CTF refinement, the Delta S-open and S-transition datasets were independently loaded into cryoSPARC v3.2.0[58] and refined to 3.1 Å and 3.4 Å resolution, respectively, using Non-uniform refinement. The overall resolution was determined based on the gold-standard criterion using a Fourier shell correlation (FSC) of 0.143. The two maps were post-processed through deepEMhancer[64]. Moreover, we performed 3D Variability analysis (3DVA) on the Delta S-open dataset in cryoSPARC to capture its continuous conformational dynamics. For the Delta S-ACE2 dataset (Supplementary Fig. 3), overall similar data processing procedure was adapted as described above for the Delta S dataset. Here, after obtaining a 3.2-Å-resolution map of S-ACE2 from 196,687 particles, we performed further local refinement on the RBD-1-ACE2 region in cryoSPARC to acquire a 3.4-Å-resolution map of this region, and 3DVA in cryoSPARC on this S-ACE2 dataset.

For the Delta S-8D3 dataset (Supplementary Fig. 5), similar data processing procedure was adapted as described for the Delta S dataset to obtain a 3.1-Å-resolution S-8D3 map. In addition, we subtracted the relatively dynamic RBD-1-8D3 region and performed one round of 3D classification, leading to a dataset of 179,073 particles, which was further refined to a 3.6-Å-resolution map of the RBD-1-8D3 region.

**Atomic model building**. To build an atomic model for the Delta S-open structure, we used the available atomic model of SARS-CoV-2 S-open (PDB 7DK3)[8] as initial model. We first fit the model into our Delta S-open map in Chimera by rigid body fitting, and manually substituted the mutations of the Delta variants in COOT[65]. We then flexibly refined the model against the density map using Rosetta[66], and finally used the phenix.real_space_refine module in Phenix for the S trimer model refinement against the map[67]. For the S-transition model, we utilized the available model SARS-CoV-2 S-transition (PDB 7KRS)[8,49] as initial template, and followed similar procedure described above for S-open model refinement. For the Delta S-ACE2 structure and the local refined RBD-1-ACE2 structure, we used the SARS-CoV-2 S-ACE2 model (PDB 7DF4)[8] as initial template, and followed similar procedure described above for model refinement. For the Delta S-8D3 structure and the local refined RBD-1-8D3 structure, we first built the homology model of the 8D3 Fab through the SWISS-MODEL server[68] by utilizing the antibody G196 (PDB 5H2B)[69] as a template. The other steps of model refinement were performed in the same way as described above. The final atomic models were validated using Phenix.molprobity command in Phenix. Interaction surface analysis was conducted by utilizing PISA server[70].

UCSF Chimera and ChimeraX were applied for figure generation, rotation measurement and coulombic potential surface analysis[71,72].

**Reporting summary**. Further information on research design is available in the Nature Research Reporting Summary linked to this article.

## Data availability

All data presented in this study are available within the figures and in the Supplementary Information, and are available from the corresponding authors upon reasonable request. For the SARS-CoV-2 Delta variant, related cryo-EM maps have been deposited at the Electron Microscopy Data Bank with accession codes EMD-32359, EMD-32360, EMD-32361, EMD-32362, EMD-32363, EMD-32364, EMD-32365, EMD-32366, and EMD-32367, and associated atomic models have been deposited in the Protein Data Bank with accession codes 7W92, 7W94, 7W98, 7W99, 7W9B, 7W9C, 7W9E, 7W9F and 7W9I for S-open, S-transition, S-ACE2-C1, S-ACE2-C2a, S-ACE2-C2b, S-ACE2-C3, S-8D3, RBD-1-8D3, and RBD-1-ACE2, respectively. Previously released structural data used during the course of this study available through the PDB: 7DCC, 7DK3, 7DK4, 7KRR, 7KRS, 7KRQ, 7DF4, 5H2B. Source data are provided with this paper.

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

## Acknowledgements

We are grateful to the staffs of the NCPSS Electron Microscopy facility, Database and Computing facility, and Protein Expression and Purification facility for instrument support and technical assistance. This work was supported by grants from the Strategic Priority Research Program of CAS (XDB37040103 and XDB29040300), National Key R&D Program of China (2017YFA0503503 and 2020YFC0845900), the NSFC (32130056 and 31872714), the NSFC-ISF 31861143028, Shanghai Academic Research Leader (20XD1404200), and the CAS Facility-based Open Research Program. This project is part of the European Union's Horizon 2020 research and innovation programme under grant agreement No 101003589 to Z.H. Chao Zhang is supported by the Youth Innovation Promotion Association of the Chinese Academy of Sciences (CAS) and Shanghai Rising-Star Program (21QA1410000).

## Author contributions

Y.C., Z.H., Y-F.W., and C.Z. designed the experiments; Y.-X.W. and C.Z. expressed and purified the proteins with assistance of Z.L., S.X. and Y.Y.; Q.H. performed cryo-EM data acquisitions; Y.-F.W. performed cryo-EM reconstructions; C.L. performed model buildings; C.Z. and Y.-X.W. performed biochemical analyses; Y.C., Z.H., Y.-F.W., C.L., and C.Z. analyzed the data; Y.C., Z.H., Y.-F.W., and Y.-X.W. wrote the manuscript with inputs from C.Z. and C.L.

## Competing interests

Z.H. and C.Z. are listed as inventors on a pending patent application for MAb 8D3. The other authors declare no competing interests.
