## [Peer Review File · Nature Communications]

Structural basis for SARS-CoV-2 Delta variant recognition of ACE2 receptor and broadly neutralizing antibodiesReviewers' Comments:

Reviewer #1:

Remarks to the Author:

The manuscript submitted by Wang et. al. attempts to provide a structural basis for the observed increases in transmissibility, infectivity and alteration of immune sensitivity in the delta variant of SARS-CoV-2. This is achieved through extensive characterisation of the delta S-trimer and its interaction with both ACE2 and neutralising antibodies through cryo-electron microscopy. Observations of the altered RBD positioning and propensity for open and closed states as compared with the WT spike is noteworthy, though direct comparison with other variants would benefit this analysis.

While the conformational flexibility of Delta S trimer has been touched upon in prior work (Yang TJ et al. Structure-activity relationships of B.1.617 and other SARS-CoV-2 spike variants, 2021 Biorxiv. Zhang J. et al. Membrane fusion and immune evasion by the spike protein of SARS-CoV-2 Delta variant, 2021 Science) the use of 3D variability analysis to explore this is of particular interest.

The authors further explore differences at the RBD-ACE2 interface between WT SARS-CoV-2 RBD and the delta variant and how this may lead to enhanced binding. The demonstration of antibody recognition of the delta variant RBDs by neutralising antibodies, some of which lose recognition of the beta variant, supports a body of literature showing enhanced immune escape of SARS-CoV-2 beta. Overall this is a compelling narrative that provides novel insight into mechanisms that potentially drive differences in infectivity and transmission between WT SARS-CoV-2 and the delta variant of concern.

Suggested Improvements

Line 127: Could the authors include a brief comparison of the delta S trimer conformations found here with those reported in Yang TJ et al. 2021, Biorxiv and Zhang J et al. 2021 Science.

Figure 2J: As the buried surface area is given in the text, could the authors give a footprint figure for the WT RBD interactions on the ACE2 surface and compare with delta (as they have done for the delta RBD in 2I).

Lines 190-208: Could the authors please include a brief comparison of the range of motion of the WT S-Trimer seen in the Xu et. al. 2021 study and the motions of the Delta S-trimer described here.

Line 218-220, figure 4A: Could the authors please provide the neutralisation values as well as fold change in figure 4A and include these values in molar concentration units for easy comparison with the field.

Line 368: Please provide a brief description of the WT-S expression protocol in addition to a paper reference.

Line 386: Please expand the protocol for the RBD expression and purification to include a similar level of detail to that given for the SARS-CoV-2 delta spike in lines 374-381.

Line 400, Figure 2A: Please specify the fitting used to obtain KD values for the BLI data.

Line 488: Please include the PDB ID of any templates used for the 8D3 Fab homology model preparation in this section.

Table S1: Please include the following values in the Cryo-EM data collection and refinement statistics:

- FSC threshold used for the map final resolution
- Resolution range of the maps.

- Initial model ID used for Atomic modelling
- Model composition including number of non-H atoms, residues and ligands
- Average, minimum and maximum B factors for proteins and ligands
- Validation statistics including Molprobity score, clashscore

Reviewer #2:

Remarks to the Author:

Summary

The SARS-CoV-2 Delta variant has become globally dominant and is causing a new wave of infections in many countries, mostly affecting unvaccinated individuals. Detailed structural and biochemical studies of spike proteins from new emerging variants is of great importance. These studies provide insights into the molecular basis for increased infectivity and reduced susceptibility to monoclonal and vaccine induced antibodies. The manuscript by Wang et al provides a comprehensive structural study of the Delta variant spike, describing several cryo-EM structures of the apo ectodomain and it's complexes with ACE2 and a neutralising antibody, 8D3. They also complement their structural data with biolayer interferometry, ELISA and pseudovirus neutralisation experiments. Collectively, these data suggest that the increased infectivity of the Delta variant spike is due to modestly increased ACE2 binding and increased sampling of the open conformation. The manuscript is reasonably well written and, as far as I can tell, the structural and functional work has been performed to a high standard. The results are relevant to the ongoing COVID-19 pandemic but the novelty of this work is diminished by several bioRxiv preprints describing similar studies (McCallum et al. 2021, Yang et al. 2021 and Saville et al. 2021) and, most recently, a peer reviewed article in Science (Zhang et al. 2021). Collectively, these articles cover most of the observations described in the present study, except for the 8D3 structure. If only the peer-reviewed article is taken into consideration, the manuscript by Wang et al provides additional structural insights into ACE2 binding and antibody mediated neutralisation.

Comments and suggestions for improvement

1. This manuscript makes several comparisons to the authors previous work published in Science Advances (Xu et al. 2021). Notably, one of the key conclusions of the present study is that the "apo" Delta-S has an open-closed ration of 75.3-24.7% compared to 6-94% in the WT spike. The authors refer to the Delta S closed conformation as "S-transition" instead of "closed", the latter of which is more consistent with the accepted nomenclature. Presumably this is done because of the small differences they describe between the WT "closed" and Delta "transition". However, the authors have overlooked a key variable which affects the comparisons of the structures and ratio of open-closed spike – the presence or absence of linoleic acid! I checked the authors previously published closed cryo-EM map (EMDB-30660) and model (PDB-7DF3). As I suspected, their previous structure contains unmodelled density for the free fatty acid. As described by Toelzer et al. 2020, linoleic acid binding leads to predominantly closed spikes that have more compacted RBDs compared to the true apo counterparts. The authors should check their Delta-S closed structures and see if the linoleic acid is present. My suspicion is that the conformational differences they see in the closed structure, compare to the WT, are due to the absence of linoleic acid in the Delta-S structure. This issue needs to be addressed throughout the results and discussion because it may affect the some of the conclusions of the paper. A more compressive comparison of open-closed ratios throughout the literature would further strengthen the authors conclusions.

2. In lines 163-167, the authors suggest that Delta-S has an increased propensity to engage ACE2 compared to the WT. However, related to my previous point, the WT spike was driven more towards the closed conformation due to linoleic acid binding. If the same is not true for Delta-S, is it a fair comparison? Was the molar excess of ACE2 the same? Was the incubation time the same? These

points need to be discussed.

3. These studies are performed on prefusion stabilised ectodomains and the results may not fully reflect what is seen in the full-length, membrane embedded spike. A single sentence caveat should be included in the discussion.

Minor points

1. The title should be "Structural basis for..."
2. Lines 23-24 should be phrased better.
3. As mentioned before, "S-transition" should just be "closed".
4. Line 690 – "are shown in light green".
5. The choice of C1, C2a, C2b and C3 to describe the ACE-bound states may cause confusion because it could also be describing the symmetry imposed during refinement.

References

- 1) McCallum et al. bioRxiv 2021.08.11.455956; doi: <https://doi.org/10.1101/2021.08.11.455956>
- 2) Yang et al. bioRxiv 2021.09.12.459978; doi: <https://doi.org/10.1101/2021.09.12.459978>
- 3) Saville et al. bioRxiv 2021.09.02.458774; doi: <https://doi.org/10.1101/2021.09.02.458774>
- 3) B. Chen et al., Science 10.1126/science.abl9463 (2021).
- 4) Xu et al, Science Advances (2021) DOI: 10.1126/sciadv.abe5575
- 5) Toelzer etl al, Science (2020) DOI: 10.1126/science.abd3255

REVIEWER COMMENTS

Reviewer #1

The manuscript submitted by Wang et. al. attempts to provide a structural basis for the observed increases in transmissibility, infectivity and alteration of immune sensitivity in the delta variant of SARS-CoV-2. This is achieved through extensive characterization of the delta S-trimer and its interaction with both ACE2 and neutralizing antibodies through cryo-electron microscopy. Observations of the altered RBD positioning and propensity for open and closed states as compared with the WT spike is noteworthy, though direct comparison with other variants would benefit this analysis.

While the conformational flexibility of Delta S trimer has been touched upon in prior work (Yang TJ et al. Structure-activity relationships of B.1.617 and other SARS-CoV-2 spike variants, 2021 BioRxiv. Zhang J. et al. Membrane fusion and immune evasion by the spike protein of SARS-CoV-2 Delta variant, 2021 Science) the use of 3D variability analysis to explore this is of particular interest.

The authors further explore differences at the RBD-ACE2 interface between WT SARS-CoV-2 RBD and the delta variant and how this may lead to enhanced binding. The demonstration of antibody recognition of the delta variant RBDs by neutralizing antibodies, some of which lose recognition of the beta variant, supports a body of literature showing enhanced immune escape of SARS-CoV-2 beta. Overall, this is a compelling narrative that provides novel insight into mechanisms that potentially drive differences in infectivity and transmission between WT SARS-CoV-2 and the delta variant of concern. The authors have addressed the points raised previously. The current manuscript is suitable for acceptance.

--Thanks for the encourage comments and insightful suggestions from our reviewer.

Suggested Improvements

Q1-1. Line 127: Could the authors include a brief comparison of the delta S trimer conformations found here with those reported in Yang TJ et al. 2021, Biorxiv and Zhang J et al. 2021 Science.

A1-1: Thanks for the suggestion. The Science paper by Zhang J et al. 2021¹ reported that the Delta S trimer exhibits three conformational states, namely S-open-1, S-open-2, and S-closed. Conformational comparison suggests that overall our S-open appears more comparable to their S-open-2 state, with the RBD-1 in our S-open lifted

up slightly (Fig. R1A); while relative to their S-open-1, our S-open appears less compact with the NTDs untwisted/outward tilted a little (Fig. R1B). Moreover, our Delta S-transition is more untwisted/open (up to 6.5°), with the RBD-1 tilted upwards slightly, than that of their S-closed (PDB ID: 7SBK) (Fig. R1C), suggesting they are in distinct state. Regarding the Delta structures from the bioRxiv paper by Yang TJ et al. 2021², our S-open exhibits similar confirmation to one of their higher populated one-RBD-up S-open states (7V7Q, Fig. R1D). We have now included this comparison in our revised manuscript (Line 139-146 on Page 5) and added Fig. R1 as Fig. S2F, H.

Fig. R1 Structural comparison between our Delta S-open or S-transition with available Delta S trimer structures. (A-B) Overlaid structural comparison between our Delta S-open (violet red) and the Delta S-open-2 (7SBO¹, dodger blue, A), or between our Delta S-open and the S-open-1 (7SBL¹, light sea green, B). (C) Structural comparison between our Delta S-transition (salmon) and the Delta S-closed (7SBK¹, light blue), indicating a clockwise rotation/untwist of our Delta S-transition relative to the Delta S-closed (left). Side view of the overlaid RBD-1 between the two structures (right), showing that the RBD-1 of our Delta S-transition is slightly lifted for 3.5° relative that of their Delta S-close¹. (D) Structural comparison between our Delta S-open and the one RBD-up S-open conformation (yellow, 7V7Q) from Yang TJ et al. 2021².

Q1-2. Figure 2J: As the buried surface area is given in the text, could the authors give a footprint figure for the WT RBD interactions on the ACE2 surface and compare with delta (as they have done for the delta RBD in 2I).

A1-2: The point is well taken. We have now added the footprint figure for the WT RBD interactions on the ACE2 surface and compared it with that of the delta RBD on ACE2

(Fig. R2). This analysis suggests that the footprint for the Delta RBD interactions on the ACE2 surface is subtly different from that of WT. We have now included Fig. R2 as Fig. S4E (Line 202 on Page 7).

Fig. R2 The footprint (in violet red) for the Delta RBD (left) and WT RBD (6M0J³, right) interactions on the ACE2 surface, with residues in proximity to RBD-1 (< 4 Å) indicated.

Q1-3. Lines 190-208: Could the authors please include a brief comparison of the range of motion of the WT S-Trimer seen in the Xu et. al. 2021 study and the motions of the Delta S-trimer described here.

A1-3: In the Xu et. al. 2021 study⁴, we applied multibody refinement through Relion software⁵ to determine the conformational dynamics of the WT S-ACE2 complex focusing only on the RBD-1-ACE2 region (Fig. R3A). In our current study, we applied 3DVA through cryoSPARC⁶ to decipher the globe continuous motion of the entire Delta S-ACE2 complex, depicting motions for not only the RBD-1-ACE2 region, but also RBD-2/3 and the coordinated NTD regions (Fig. R3B). We thus compared motions only for the RBD-1-ACE2 region. In general, the WT RBD-1-ACE2 motions displayed in eigenvectors 1 and 3 are comparable to the Delta RBD-1-ACE2 movements exhibited in motion 1 and 3, respectively, in terms of direction and range. Still, the eigenvector 2 of the WT displays a distinct RBD-1-ACE2 swing motion (towards RBD-3) relative to the motion 2 of Delta, which displays a RBD-1-ACE2 movement towards RBD-2. Since the computational software/methods and focused regions are not the same between the two systems, we prefer not to include this comparison in the manuscript.

Fig. R3 Conformational dynamics of the SARS-CoV-2 WT S-ACE2 and the Delta S-ACE2 complexes. (A) Conformational dynamics of the WT S-ACE2 focused only on the RBD-1-ACE2 region (taken from Xu et. al. Sci Adv. 2021)⁴. The swing angular range and direction are indicated by dark red arrow. (B) Conformational dynamics of the Delta S-ACE2 complex. The swing angular range and direction are indicated by black arrow.

Q1-4. Line 218-220, figure 4A: Could the authors please provide the neutralization values as well as fold change in figure 4A and include these values in molar concentration units for easy comparison with the field.

A1-4: We thank the reviewer for this constructive comment. As suggested, we have now included the neutralization values in the revised Fig. 4A (also shown below for convenient view).

MAb	IC50 value (ng/mL)			IC50 value (nM)			IC50 fold change in variant PV neutralization (relative to WT)		
	Beta	Kappa	Delta	Beta	Kappa	Delta	Beta	Kappa	Delta
3C1	1460	338.2	417.5	9.733	2.255	2.783	-2.9	1.5	1.2
2H2	>10000	>10000	3.2	>66.667	>66.667	0.021	<-400	<-400	7.8
2G3	3.9	1.4	83.7	0.026	0.009	0.558	1.8	5.0	-12
3A2	>10000	>10000	31.4	>66.667	>66.667	0.209	<-204	<-204	1.6
8D3	6.3	2.2	7.6	0.042	0.015	0.051	1.1	3.2	-1.1

Fig. 4 Neutralization breadth and binding properties of the MAbs (3C1, 2H2, 2G3, 3A2, and 8D3) against SARS-CoV-2 variants. (A) Neutralization values and fold change in neutralization IC50 of the MAbs against the variant pseudoviruses, relative to the WT pseudovirus. A minus sign (-) denotes decrease. Orange shade, more than 10-fold decrease; red shade, more than 100-fold decrease.

Q1-5. Line 368: Please provide a brief description of the WT-S expression protocol in addition to a paper reference.

A1-5: The suggestion is well taken. A brief description of the WT-S expression protocol is now included in the Method section of the revised manuscript (please see Page 14, Line 389-392), to read “Briefly, the prefusion-stabilized S ectodomain (residues M1–Q1208) of SARS-CoV-2 Wuhan-Hu-1 strain (GenBank ID: MN908947.3) was produced in the transfected HEK293F cells and then purified from the culture supernatant using Ni-NTA affinity resin.”

Q1-6. Line 386: Please expand the protocol for the RBD expression and purification to include a similar level of detail to that given for the SARS-CoV-2 delta spike in lines 374-381.

A1-6: As suggested, the detailed description of mutant RBD expression and purification protocol has now been provided in the Methods section of the revised manuscript (please see Page 14, Line 409-415), to read “The resulting mutant plasmids were separately transfected into HEK293F cells using PEI and cultured for 5 days. The culture supernatants were then collected by centrifugation and loaded onto the pre-equilibrated Ni-NTA affinity column. After wash, the bound His-tagged mutant RBD proteins were eluted with elution buffer (20 mM Tris-HCl pH 7.9, 500 mM NaCl, 250 mM imidazole). The eluted proteins were analyzed by SDS-PAGE and dialyzed against PBS, and protein concentration was determined by Bradford method.”

Q1-7. Line 400, Figure 2A: Please specify the fitting used to obtain KD values for the BLI data.

A1-7: Thanks for the suggestion. We have now added the fitting curve in Fig. 2A (black lines) and specified the fitting methods in our revised manuscript in Method section (please see Page 15, Line 428-429), to read “The data were corrected by subtracting reference sample and then fitted to a 1:1 binding model for determination of affinity constants using the software Octet Data Analysis 11.0”. For the convenience of our reviewers and editor, we also show the revised Fig. 2A below.

Fig. 2A Measurement of the binding affinity between ACE2 and the S trimer of the Delta (left) or G614 (right) variants using bio-layer interferometry (BLI). ACE2 concentrations tested were shown. Raw sensor grams and fitting curves were shown in color and black, respectively.

Q1-8. Line 488: Please include the PDB ID of any templates used for the 8D3 Fab homology model preparation in this section.

A1-8: We have followed the suggestion to added the related template used for 8D3 homology model building in method, to read “We first built the homology model of the 8D3 Fab through the SWISS-MODEL server by utilizing the antibody G196 (PDB: 5H2B)⁷ as a template.” (Line 517-518 on Page 18)

Q1-9. Table S1: Please include the following values in the Cryo-EM data collection and refinement statistics:

- FSC threshold used for the map final resolution
- Resolution range of the maps.
- Initial model ID used for Atomic modelling
- Model composition including number of non-H atoms, residues and ligands
- Average, minimum and maximum B factors for proteins and ligands
- Validation statistics including Molprobity score, clashscore

A1-9: The points are well taken and we have now added the suggested values in Table S1, which is also shown below for convenient view.

Table S1. Cryo-EM data collection and refinement statistics for Delta S, Delta S-ACE2, and Delta S-8D3

	Delta S	Delta S-ACE2						Delta S-8D3	
Data collection									
EM equipment	Titan Krios		Titan Krios					Titan Krios	
Voltage (kV)	300		300					300	
Detector	Gatan K3 camera		Gatan K3 camera					Gatan K3 camera	
Pixel size (Å)	0.893		1.093					1.093	
Electron dose (e ⁻ /Å ²)	50.2		50.2					50.2	
Exposure time (s)	2		3					3	
Frames	40		30					30	
Defocus range (µm)	-0.8 to -2.5		-0.8 to -2.5					-0.8 to -2.5	
Reconstruction									
Softwares	Relion 3.1&cryoSPARC								
Structures	S-open	S-transition	C1	C2a	C2b	C3	RBD-1-ACE2	S-8D3	RBD-1-8D3
Final particles	103,096	33,761	23,365	39,681	41,938	91,703	196,687	246,240	179,073
Symmetry	C1	C1	C1	C1	C1	C1	C1	C1	C1
FSC threshold	0.143								
Final overall resolution (Å)	3.1	3.4	3.6	3.4	3.4	3.2	3.4	3.1	3.6
Resolution Range (Å)	2.6-6.6	2.9-6.9	3.1-7.5	3.1-7.5	3.1-7.5	2.8-6.8	2.9-3.7	2.6-8.6	3.5-4.7
Atomic modeling									
Softwares	Rosetta & Phenix & Coot								
Initial model ID	7DK3	7KRS	7DF4	7DF4	7DF4	7DF4	7DF4	7DK3; 5H2B	7DK3; 5H2B
Averaged Bfactor	88.9	112.3	183.8	144.8	150.8	71.4	43.7	89.3	82.6
number of non-H atoms, residues and ligands	25,074; 3,206; 0	25,085; 3,206; 0	29,741; 3,777; 0	29,741; 3,777; 0	29,741; 3,777; 0	29,741; 3,777; 0	6,413; 791; 0	28,186; 3,611; 0	4,858; 625; 0
Rms deviations									
Bond length (Å)	0.0066	0.0042	0.0043	0.0045	0.0045	0.0048	0.0053	0.0046	0.0044
Bond Angle (°)	1.09	1.06	0.98	0.99	0.99	1.01	1.12	1.01	1.10
Ramachandran plot (%)									
Favored	95.85	95.76	97.40	96.95	97.16	97.21	95.17	96.07	96.28
Allowed	4.15	4.24	2.57	2.93	2.81	2.79	4.83	3.93	3.72
Outliers	0.00	0.00	0.03	0.12	0.03	0.00	0.00	0.00	0.00
Molprobrity score	1.45	1.46	1.02	1.10	1.08	1.08	1.51	1.37	1.38
Clash score	3.69	3.81	1.60	1.65	1.76	1.76	3.91	3.14	3.37

Reviewer #2

The SARS-CoV-2 Delta variant has become globally dominant and is causing a new wave of infections in many countries, mostly affecting unvaccinated individuals. Detailed structural and biochemical studies of spike proteins from new emerging variants is of great importance. These studies provide insights into the molecular basis for increased infectivity and reduced susceptibility to monoclonal and vaccine induced antibodies. The manuscript by Wang et al provides a comprehensive structural study of the Delta variant spike, describing several cryo-EM structures of the apo ectodomain and its complexes with ACE2 and a neutralising antibody, 8D3. They also complement their structural data with biolayer interferometry, ELISA and pseudovirus neutralisation experiments. Collectively, these data suggest that the increased infectivity of the Delta variant spike is due to modestly increased ACE2 binding and increased sampling of the open conformation. The manuscript is reasonably well written and, as far as I can tell, the structural and functional work has been performed to a high standard. The results are relevant to the ongoing COVID-19 pandemic but the novelty of this work is diminished by several bioRxiv preprints describing similar studies (McCallum et al. 2021, Yang et al. 2021 and Saville et al. 2021) and, most recently, a peer reviewed article in Science (Zhang et al. 2021). Collectively, these articles cover most of the observations described in the present study, except for the 8D3 structure. If only the peer-reviewed article is taken into consideration, the manuscript by Wang et al provides additional structural insights into ACE2 binding and antibody mediated neutralisation.

--We appreciate the encouraging comments and suggestions from this reviewer, which led us to explore more thoroughly to improve our manuscript.

Major comments:

Q2-1. This manuscript makes several comparisons to the authors previous work published in Science Advances (Xu et al. 2021). Notably, one of the key conclusions of the present study is that the “apo” Delta-S has an open-closed ration of 75.3-24.7% compared to 6-94% in the WT spike. The authors refer to the Delta S closed conformation as “S-transition” instead of “closed”, the latter of which is more consistent with the accepted nomenclature. Presumably this is done because of the small differences they describe between the WT “closed” and Delta “transition”. However, the authors have overlooked a key variable which affects the comparisons of the

structures and ratio of open-closed spike – the presence or absence of linoleic acid! I checked the authors previously published closed cryo-EM map (EMDB-30660) and model (PDB-7DF3). As I suspected, their previous structure contains unmodelled density for the free fatty acid. As described by Toelzer et al. 2020, linoleic acid binding leads to predominantly closed spikes that have more compacted RBDs compared to the true apo counterparts. The authors should check their Delta-S closed structures and see if the linoleic acid is present. My suspicion is that the conformational differences they see in the closed structure, compare to the WT, are due to the absence of linoleic acid in the Delta-S structure. This issue needs to be addressed throughout the results and discussion because it may affect the some of the conclusions of the paper. A more compressive comparison of open-closed ratios throughout the literature would further strengthen the authors conclusions.

A2-1: Thanks for the insightful comments from our reviewer. We would like to point out that we judged our Delta S-transition state by comparing our structure with the parental strain D614G (termed G614) S-closed structure (PDB ID:7KRQ) from Bing Chen group⁸, but not with our previous WT S-closed structure⁴. Neither our Delta S-transition nor their G614 S-closed maps contain engaged linoleic acid (Fig. R5A, H), hence, the conformational differences between them are not related to linoleic acid. Furthermore, as described in A1-1, compared with the recently reported Delta S-closed state¹ which also lacks linoleic acid, our Delta S-transition appears more untwisted/open (up to 6.5°) with the RBD-1 slightly lifted up (Fig. R1C), indicating that it is in a distinct conformation from their S-closed. Moreover, the three protomers in our Delta S-transition appear asymmetric with a gap emerging between RBD-1 and -3, which is not the case between RBD-2 and -3 (Fig. R4A). Also, the “630 loops” and fusion peptides in our S-transition are partially or fully disordered (Fig. 4B), which is usually the case in the transition or open state of WT/G641/Delta spike^{1,8,9}. Collectively, these features suggest that this S-transition is indeed not in the closed state. Still, the upward tilting of RBD-1 in this structure is not that dramatic relative to the closed state, we therefore interpret it as the initial stage of transition towards the open state. We have now coordinated the above descriptions in our revised manuscript (Line 134-139 on Page 5), and included Fig. R4 as Fig. S2G, I.

Fig. R4 Structural features of our Delta S-transition. (A) Top view of our Delta S-transition cryo-EM map, which appears asymmetric with a gap emerging between RBD-1 and -3 (indicated by dotted blue oval), which is not the case between RBD-2 and -3. (B) In the Delta S-transition, all three FPs (blue) are disordered, and all the 630 loops (red) are partially disordered.

The reviewer is right that there is linoleic acid (termed LA afterwards) in our previous WT closed state structure (EMDB-30660, Fig. R5B) (Xu *et al.*, 2021, *Sci. Adv.*⁴), which is also true for other tightly closed WT S trimer structures (6ZB5, 6ZGE, and 6XR8) from different groups (Toelzer *et al.* 2020, *Science*.¹⁰; Wrobel, *et al.*, 2020, *Nat Struct Mol Biol*.¹¹; Cai *et al.*, 2020, *Science*.⁹) (Fig. R5C-E, Table R1). We agree with the reviewer that “As described by Toelzer *et al.* 2020, linoleic acid binding leads to predominantly closed spikes that have more compacted RBDs compared to the true apo counterparts”. We thus surveyed related literatures^{1,4,8-13} to perform a more comprehensive comparison of open-closed ratios, including our recent work on SARS-CoV-2 Kappa and Beta variants¹⁴ (Table R1). Here, the Delta S trimer is obtained in the same construction and purification condition to that of our Kappa and Beta S¹⁴, and all of them have no engaged LA. Even so, the Delta S exhibits a higher open-transition ratio (75.3%-24.7%) than that of the Kappa and Beta S (both around 50%-50%)¹⁴, substantiating our notion that Delta S displays a population shift towards the open state. Furthermore, the population of our Delta S-open (up to 75%) is comparable to the recent reported Delta S (~70%, Bing Chen, 2021, *Science*.¹), also higher than the other S trimers in the absence of LA (ranging from 32% to 50%) (Walls *et al.*, 2020, *Cell*.¹²; Zhang *et al.*, 2021, *Science*.⁸), except for one case with all the WT S in the open state (Wrapp *et al.*, 2020, *Science*.¹³). Thus, we replaced the population comparison with our previous WT closed S with that of our recent Kappa and Beta S trimers¹⁴, and also presented the comparison with the other spikes without engaged LA in our revised manuscript (Line 109-112, Line 114-117 on Page 5, and Line 313-315 on Page 11). Nonetheless, the conclusion about open-closed ratio and that the Delta S displays a population shift towards the open state is retained.

Fig. R5 LA occupancy status of SARS-CoV-2 S trimers. (A-B) The LA (in red stick) occupancy status of our Delta S-transition (current study) (A) and the WT S-closed state (PDB: 7DF3, our previous study⁴) (B). It appears there is no LA density in all three RBDs in the Delta S-transition, and an additional LA density (indicated by dotted red oval) in each RBD in our previous WT S-close structure. (C-E) Three other previously reported tightly closed WT S structures (6ZB5¹⁰, 6ZGE¹¹, and 6XR8⁹) appear to contain LA density in the RBDs. (F-I) There is no LA density in the other previous S-closed structures, including two WT S-closed structures (PDB: 6VSB¹³, 6VXX¹²) (F-G) and two variants, i.e. G614 S-closed (PDB: 7KRQ)⁸ (H) and Delta S-closed structures (PDB: 7SBK¹) (I).

Table R1 Open-closed ratios in different SARS-CoV-2 spike structural research

Reference	SARS-CoV-2 strain	Linoleic acid (Y/N)	Open vs. closed
This study	Delta	N	75.3-24.7% (open-transition)
Our Beta ¹⁴	Beta	N	53.1-46.9% (open-transition)
Our Kappa ¹⁴	Kappa	N	49.1-50.9% (open-transition)
Xu et al. , 2021, Sci. Adv. ⁴	WT	Y	6-94%
Toelzer et al. 2020, Science . ¹⁰	WT	Y	30-70%
Cai et al. , 2020, Science . ⁹	WT	Y	34-66%
Wrobel, et al. , 2020, Nat Struct Mol Biol. ¹¹	WT	Y	17-83%

Walls et al., 2020, Cell. ¹²	WT	N	~50-50%
Wrapp et al., 2020, Science. ¹³	WT	N	All open
Zhang et al., 2021, Science. ⁸	G614	N	32-36-32% (open-intermediate-closed)
Zhang et al., 2021, Science. ¹	Delta	N	70-30%

Q2-2. In lines 163-167, the authors suggest that Delta-S has an increased propensity to engage ACE2 compared to the WT. However, related to my previous point, the WT spike was driven more towards the closed conformation due to linoleic acid binding. If the same is not true for Delta-S, is it a fair comparison? Was the molar excess of ACE2 the same? Was the incubation time the same? These points need to be discussed.

A2-2: The point is well taken and we have removed the population comparison with the WT S-ACE2 complex. Still, as we point out in A2-1, in our recent study on the Beta and Kappa S trimer and their complexes with ACE2, we constructed the Beta and Kappa S in the same way as the Delta S, and there is no engaged linoleic acid in the Beta and Kappa S trimer¹⁴. Also, we assembled their complex with ACE2 in exactly the same condition, i.e. S to ACE2 molar ratio at 1:4 and incubated on ice for 20 min, which justifies the population distribution comparison of Delta S-ACE2 with that of Beta or Kappa S-ACE2 complexes. We have now modified the description in lines 163-167 as the following, “Noteworthy, compared with the Beta or Kappa S-ACE2 complexes from our recent study (the three-RBD-up C3 conformation is at 27.7% and 34.1%, respectively, purified and assembled in the same condition as for the Delta variant)¹⁴, the Delta S-ACE2 complex showed a considerable population shift towards the more ACE2 engaged C3 configuration (46.6% three RBD-up C3 state, Fig. 2C), which could be beneficial for subsequent S1 shedding and the S trimer transition towards postfusion state”. (Now at Line 182-186 on Page 7).

Q2-3. These studies are performed on prefusion stabilised ectodomains and the results may not fully reflect what is seen in the full-length, membrane embedded spike. A single sentence caveat should be included in the discussion.

A2-3: Thanks for the comments. Yes, our previous and current studies are performed on prefusion stabilized ectodomains. However, our structures are quite similar to those obtained with full-length, membrane embedded spike. For example, the structure of closed state WT S trimer from our previous work⁴ share similar conformation with that of the WT full length S trimer (also show it below as Fig. R6A), and the conformation of our current Delta S-open state is similar to that of the full-length Delta S-open-2 state published recently¹ (Fig. R6B). Nonetheless, it is still possible that the results generated on prefusion stabilized ectodomains may not fully reflect what is seen in the full-length, membrane embedded spike. Accordingly, we have now added a statement in our revised manuscript (Line 142-144 on Page 6).

Fig. R6 Conformation comparison between our prefusion stabilized ectodomains of SARS-CoV-2 S trimer and the full-length, membrane embedded spike for WT and Delta variant. (A) Structural comparison between our previous S-closed structure⁴ (blue, PDB ID: 7DF3) and the wild-type S trimer prefusion structure (PDB: 6XR8, orange). The two structures are in similar conformation. (B) Overlaid structural comparison between our Delta S-open state with the Delta S-open2 (PDB ID: 7SBO¹) state.

Minor comments:

1. The title should be “Structural basis for...”

--The suggestion is well taken.

2. Lines 23-24 should be phrased better.

--The point is well taken and we have changed it to “Uncovering the underlying structural basis of the enhanced transmission and altered immune sensitivity of the Delta variant is particularly important.”.

3. As mentioned before, “S-transition” should just be “closed”.

--As discussed in A2-1, due to the asymmetry in S1 region and disordered 630 loops, we prefer to use "S-transition" instead of "closed". Still, we define it clearer, to call it "open initiation transition state".

4. Line 690 – "are shown in light green".

--We have followed the suggestion to change "were" to "are" in Line 690 (Now in Line 738).

5. The choice of C1, C2a, C2b and C3 to describe the ACE-bound states may cause confusion because it could also be describing the symmetry imposed during refinement.

--The letter "C" in C1, C2a, C2b and C3 means conformer or conformational state. Since S1 or S2 has been used to represent different subunits in SARS-CoV-2 spike, we therefore choose to use "C1, C2a, C2b and C3", which is also commonly used in other structural literatures¹⁴⁻¹⁶.

References:

- 1 Zhang, J. *et al.* Membrane fusion and immune evasion by the spike protein of SARS-CoV-2 Delta variant. *Science*, eabl9463, doi:10.1126/science.abl9463 (2021).
- 2 Yang, T.-J. *et al.* Structure-activity relationships of B.1.617 and other SARS-CoV-2 spike variants. *BioRxiv*, doi:10.1101/2021.09.12.459978 (2021).
- 3 Lan, J. *et al.* Structure of the SARS-CoV-2 spike receptor-binding domain bound to the ACE2 receptor. *Nature* **581**, 215-220, doi:10.1038/s41586-020-2180-5 (2020).
- 4 Xu, C. *et al.* Conformational dynamics of SARS-CoV-2 trimeric spike glycoprotein in complex with receptor ACE2 revealed by cryo-EM. *Sci Adv* **7**, doi:10.1126/sciadv.abe5575 (2021).
- 5 Fernandez-Leiro, R. & Scheres, S. H. W. A pipeline approach to single-particle processing in RELION. *Acta Crystallogr D Struct Biol* **73**, 496-502, doi:10.1107/S2059798316019276 (2017).
- 6 Punjani, A., Rubinstein, J. L., Fleet, D. J. & Brubaker, M. A. cryoSPARC: algorithms for rapid unsupervised cryo-EM structure determination. *Nat Methods* **14**, 290-296, doi:10.1038/nmeth.4169 (2017).
- 7 Tatsumi, K. *et al.* G196 epitope tag system: a novel monoclonal antibody, G196, recognizes the small, soluble peptide DLVPR with high affinity. *Sci Rep* **7**, 43480, doi:10.1038/srep43480 (2017).
- 8 Zhang, J. *et al.* Structural impact on SARS-CoV-2 spike protein by D614G substitution. *Science* **372**, 525-530, doi:10.1126/science.abf2303 (2021).
- 9 Cai, Y. *et al.* Distinct conformational states of SARS-CoV-2 spike protein. *Science*, doi:10.1126/science.abd4251 (2020).
- 10 Toelzer, C. *et al.* Free fatty acid binding pocket in the locked structure of SARS-CoV-2 spike protein. *Science* **370**, 725-730, doi:10.1126/science.abd3255 (2020).
- 11 Wrobel, A. G. *et al.* SARS-CoV-2 and bat RaTG13 spike glycoprotein structures inform on virus evolution and furin-cleavage effects. *Nature structural &*

- molecular biology* **27**, 763-767, doi:10.1038/s41594-020-0468-7 (2020).
- 12 Walls, A. C. *et al.* Structure, Function, and Antigenicity of the SARS-CoV-2 Spike Glycoprotein. *Cell* **181**, 281-292 e286, doi:10.1016/j.cell.2020.02.058 (2020).
 - 13 Wrapp, D. *et al.* Cryo-EM structure of the 2019-nCoV spike in the prefusion conformation. *Science* **367**, 1260-1263, doi:10.1126/science.abb2507 (2020).
 - 14 Wang, Y. *et al.* Conformational dynamics of the Beta and Kappa SARS-CoV-2 spike proteins and their complexes with ACE2 receptor revealed by cryo-EM. *Nat Commun* doi:10.1038/s41467-021-27350-0 (2021). (accepted)
 - 15 Yan, R. *et al.* Structural basis for the different states of the spike protein of SARS-CoV-2 in complex with ACE2. *Cell Res*, doi:10.1038/s41422-021-00490-0 (2021).
 - 16 Ding, Z. *et al.* Structural Snapshots of 26S Proteasome Reveal Tetraubiquitin-Induced Conformations. *Mol Cell* **73**, 1150-1161 e1156, doi:10.1016/j.molcel.2019.01.018 (2019).

Reviewers' Comments:

Reviewer #1:

Remarks to the Author:

The authors have gone to considerable effort to produce a very thorough response to each of the comments raised in the initial review and I am satisfied that they have addressed the issues identified during this. I am therefore happy that the work supports the conclusions drawn in the manuscript and that the data analysis and methodology are sound.

Reviewer #2:

Remarks to the Author:

The authors have addressed all my comments and the paper has been significantly improved.